# Sequential addition of cations increases photoluminescence quantum yield of metal nanoclusters near unity

Xue Wang [1,4], Yuan Zhong[1,4], Tingting Li[2], Kunyu Wang[1], Weinan Dong[1], Min Lu[1], Yu Zhang[1], Zhennan Wu [1] ✉, Aiwei Tang [3] ✉ & Xue Bai [1] ✉

Photoluminescence is one of the most intriguing properties of metal nanoclusters derived from their molecular-like electronic structure, however, achieving high photoluminescence quantum yield (PLQY) of metal core-dictated fluorescence remains a formidable challenge. Here, we report efficient suppression of the total structural vibrations and rotations, and management of the pathways and rates of the electron transfer dynamics to boost a near-unity absolute PLQY, by decorating progressive addition of cations. Specifically, with the sequential addition of $Zn^{2+}$, $Ag^+$, and $Tb^{3+}$ into the 3-mercaptopropionic acids capped Au nanoclusters (NCs), the low-frequency vibration of the metal core progressively decreases from 144.0, 55.2 to 40.0 cm$^{-1}$, and the coupling strength of electrons-high-frequency vibration related to surface motifs gradually diminishes from 40.2, 30.5 to 14.4 meV. Moreover, introducing cation additives significantly reduces electron transfer time from 40, 27 to 12 ps in the pathway from staple motifs to the metal core. This benefits from the shrinkage of the total structure that speeds up the shell-core electron transition, and in particular, the $Tb^{3+}$ provides a hopping platform for the excited electrons as their intrinsic ladder-like energy level structure. As a result, it allows a remarkable enhancement in PLQY, from 51.2%, 83.4%, up to 99.5%.

Monolayer-protected metal nanoclusters (NCs), composed of few- to hundred-metal atoms, feature discrete electronic energy levels and thus perform charming molecule-like photoluminescence[1–4]. Typically, the metal kernel-dominated fluorescence features a small Stokes shift, narrow emission band, and short decay lifetimes, finding increasing acceptance in lighting, sensing, imaging, etc[5–10]. However, it is still a big challenge at the current stage to achieve high photoluminescence quantum yield (PLQY). On the one hand, in addition to the inherent metal core vibrations, the vibrations of the interfacial staple motifs and the terminal ligand groups would lead to the total structural motions

of metal NCs with a wide range in vibrational frequencies (e.g., low-frequency breathing-/quadrupolar-like vibrations)[11]. On this basis, low- and high-frequency vibrations can be induced in the metal core and staple motifs, respectively[12,13]. The coupling of multiple vibrations and excited-state electrons can generate additional non-radiative channels eventually quenching the fluorescence of metal NCs. On the other hand, manipulating the electron transfer process from surface staple motifs with high light-absorption capability to the emission center of the metal core is a well-recognized potential strategy, yet remains challenging in tailoring the electron transfer dynamics (e.g., pathways

[1]State Key Laboratory of Integrated Optoelectronics, College of Electronic Science and Engineering, Jilin University, Changchun, P. R. China. [2]College of Materials Science and Engineering, Jilin Jianzhu University, Changchun, P. R. China. [3]Key Laboratory of Luminescence and Optical Information, Ministry of Education, School of Physical Science and Engineering, Beijing Jiaotong University, Beijing, P. R. China. [4]These authors contributed equally: Xue Wang, Yuan Zhong. ✉e-mail: wuzn@jlu.edu.cn; awtang@bjtu.edu.cn; baix@jlu.edu.cn

and rates of the radiative relaxation) due to its poor structural accessibility and functionalization[14,15]. Given these limitations, the PLQY of core-dictated photoluminescence in metal NCs suffers a low level (< 10%, in general). To date, a nontrivial undertaking remains to map out how to modulate the total structural motion and electron transfer dynamics of metal NCs, thus meeting the increasing demand for emission intensity enhancement in a colloidal state.

Cation additive engineering has been recognized as an efficient strategy for tuning the structural properties (e.g., structural motions in frequency, anisotropy, polarity, and magnitude) of various chromophores (from inorganic ions, organic molecules, metal-organic complexes, nano-/micro-crystals, to macro-solids), which in turn regulates their electronic structure and excited state electronic dynamics, further ultimately affects their luminescent properties[16–20]. In particular, metal NCs have been described as the "superatom complex" model with closed valence-electron shells (i.e., 2, 8, 18, 20, etc.)[21,22]. The electron-rich nature makes them highly susceptible to electronic/electrostatic aggression. In addition, the dynamic adsorption-desorption balance of the surface ligands, in a "divide and protect" mode, further deepens the impact on their charge-related luminescent properties. In concrete terms, (1) the alloy of heteroatoms in metal NCs can significantly modulate the structural vibrations and the electron-vibration coupling strength[23–26]. For example, the PLQY of $Ag_{29-x}Au_x$ NCs increased from 0.9% to 24% after dopped with 40 mmol% Au precursor[27]; a record-high NIR PLQY of 35% at 900 nm was also reported in $AuAg_{24}$ NCs from pristine 3.5% PLQY $Ag_{25}$ NCs[28]. (2) Adding cations can adjust the electron transfer rate and pathway during the photo-excitation of metal NCs. As a representative, the Scandium ion ($Sc^{3+}$) as a model Lewis acid allows an intramolecular charge transfer state to accelerate the photoinduced electron transfer rate from Au NCs to the minocycline molecules[29]. (3) the hierarchical structure of metal NCs can be diversified in the presence of cations. In a seminal work, Zhu et al. reported that the monodisperse $Ag_{29}$ NCs (0D) can be assembled into linear chains (1D), grid networks (2D), and superstructures (3D) with the aid of $Cs^+$ cations[30]. The emission intensity of $Ag_{29}$ NCs in 0D, 1D, and 3D enhanced for 1.7, 2.1, and 2.3 folds, respectively, compared with that of $Ag_{29}$ NCs in 2D structure due to the changed electron-vibration interactions. Despite the continued advances in the cation additive-mediated photoluminescent metal NCs, they still suffer from the ambiguous law in the influence of cation species on the PLQY, and in particular with a limited enhancement. It, therefore, studies on cation additive engineering in luminescent metal NCs deserve the necessary attention to permit maximization and even customization of their optical properties.

Herein, toward achieving near-unity PLQY of metal NCs, we elaborately design the subsequent cations additives in the metal NCs to collectively suppress the total vibrations, and direct the electron transfer channel with a regulable rate among the multiple structural domains. Remarkably, the colloidal as-added metal NCs feature a near-unity PLQY at room temperature. In detail, we employed $Zn^{2+}$, $Ag^+$, and $Tb^{3+}$ in sequence as serial cations and the 3-mercaptopropionic acids (MPA)-protected Au NCs as model hosts. Wherein, the first $Zn^{2+}$ additive drives colloidal self-assembly of Au NCs by electrostatic interactions with deprotonated carboxyl of MPA ligand, leveraging the aggregation-induced emission effect to attain the initial emission enhancement. Then, the second $Ag^+$ additive can greatly change the electronic structure and suppress the structural relaxation of Au NCs by the formation of an Ag-alloyed Au kernel. In parallel, the meanwhile incorporation of $Ag^+$ at the interfacial staple motif lengthens its length with suppressed structural motion, thus enabling a secondary enhancement in PLQY. Furthermore, the third additive of $Tb^{3+}$ ions not only reinforces the total structural rigidity but also opens up a universal law that the introduction of lanthanide cations ($Ln^{3+}$) establishes

an intermediate energy level boosting the electron hopping during the electron transfer from the surface state to the metallic core state. Finally, the triple stepwise cation additives engineering collectively confer Au NCs with near-unity PLQY, up to 99.5%. Our findings suggest a versatile sequential cations addition strategy in the design of colloidal metal NCs with intense luminescence, adding to their acceptance in diverse sectors of practical applications.

## Results

### Optical properties and structural characterization

The general synthetic scheme of Au-based NCs is displayed in Fig. 1a. In a typical synthesis, MPA ligands served as protective agents cum reductants to reduce Au(III) to Au(I). The resultant Au NCs are optical-silence to the naked eye (PLQY ≈ 0.0%, Fig. 1a and Supplementary Fig. 1a). Surprisingly when sequentially adding $Zn^{2+}$, $Ag^+$, and $Tb^{3+}$ cations to the Au NCs (denoted as **Au-Zn, Au-Zn/Ag,** and **Au-Zn/Ag/Tb** NCs hereafter), the emission peak experiences continued shift from 501 to 490 nm, and the corresponding absolute PLQYs were recorded to increase to 51.2%, 83.4%, and finally reach up to be near-unity of 99.5% (Supplementary Fig. 2). The sequential addition of cations greatly interfered with their absorption and excitation behaviors. As shown in Fig. 1b, most of the absorption of the Au NCs is contributed by the characteristic absorption transitions of the surface MPA ligands (< 300 nm). Only a tiny fraction comes from the characteristic absorption transitions of the metal core (> 400 nm), which is attributed to the petite size of the metal core in the NCs[31]. The characteristic Au $4f$ XPS spectra of Au NCs demonstrate that they are in classical Au(0)@Au(I) core-shell structure (Supplementary Fig. 1b). After adding $Zn^{2+}$, the characteristic absorption transitions of the metal core centered at 462 nm become more pronounced in **Au-Zn** NCs, indicating the growth of the metal core. Moreover, after introducing $Ag^+$ (the optimized Au/Ag molar ratio is 4:1, see Supplementary Fig. 3) into **Au-Zn** NCs, the characteristic absorption peaks of the metal core blue-shifted to 423 nm. This is because the $Ag^+$ increases the optical energy gap from 2.59 to 2.69 eV (Supplementary Fig. 4). Finally, the addition of $Tb^{3+}$ (the optimized AuAg/Tb molar ratio is 4:1, see Supplementary Fig. 5) into **Au-Zn/Ag** NCs has almost no effect on the characteristic absorption transitions of the core state, which implies that the $Tb^{3+}$ may be anchored on the surface of the **Au-Zn/Ag/Tb** NCs with the deprotonated carboxyl groups in MPA ligands. The high energy absorption band that arises between 300–400 nm is contributed by the characteristic absorption transitions of the interfacial staple motif. It is impressive that a photoinduced electron transfer (PET) band was observed spanning 250–450 nm overlapping the motif excitation peak (~ 370 nm) in the excitation spectrum of **Au-Zn, Au-Zn/Ag,** and **Au-Zn/Ag/Tb** NCs (Fig. 1c)[32–34]. Respectively, the contribution of the PET band and the excitation of staple motifs gradually increases with the subsequent addition of the metal cations. In addition, these serial NCs manifest narrow full width at half maximum (FWHM) of 86–108 meV and a small Stokes shift of 61–85 meV (Supplementary Table 1).

To study the intrinsic mechanism of emission enhancement of the serial Au-based NCs, we first investigated their structure and composition evolution during the subsequent cation additives engineering. Although the single-crystal structural anatomy is paramount in NC research, it is always challenging for the aqueous systems[35–37]. In particular in our system: there are strong electrostatic interactions between different cations and deprotonated COO⁻ groups in MPA ligands, associated with dipolar interactions between MPA ligands in water[16]; and the aliphatic alkane of MPA ligands is highly flexible, which makes it hard to induce stable inter-ligand interactions for long-range ordered structures, hence impeding the final formation of X-ray-quality crystals[38]. To push our step in addressing the composition of as-synthesized NCs, we conducted the matrix-assisted laser desorption ionization-time-of-flight mass. As shown in Supplementary Fig. 6, the mass signals at 1147.3 (**Au-Zn**) and 1277.2 (**Au-Zn/Ag,** and **Au-Zn/Ag/Tb**)

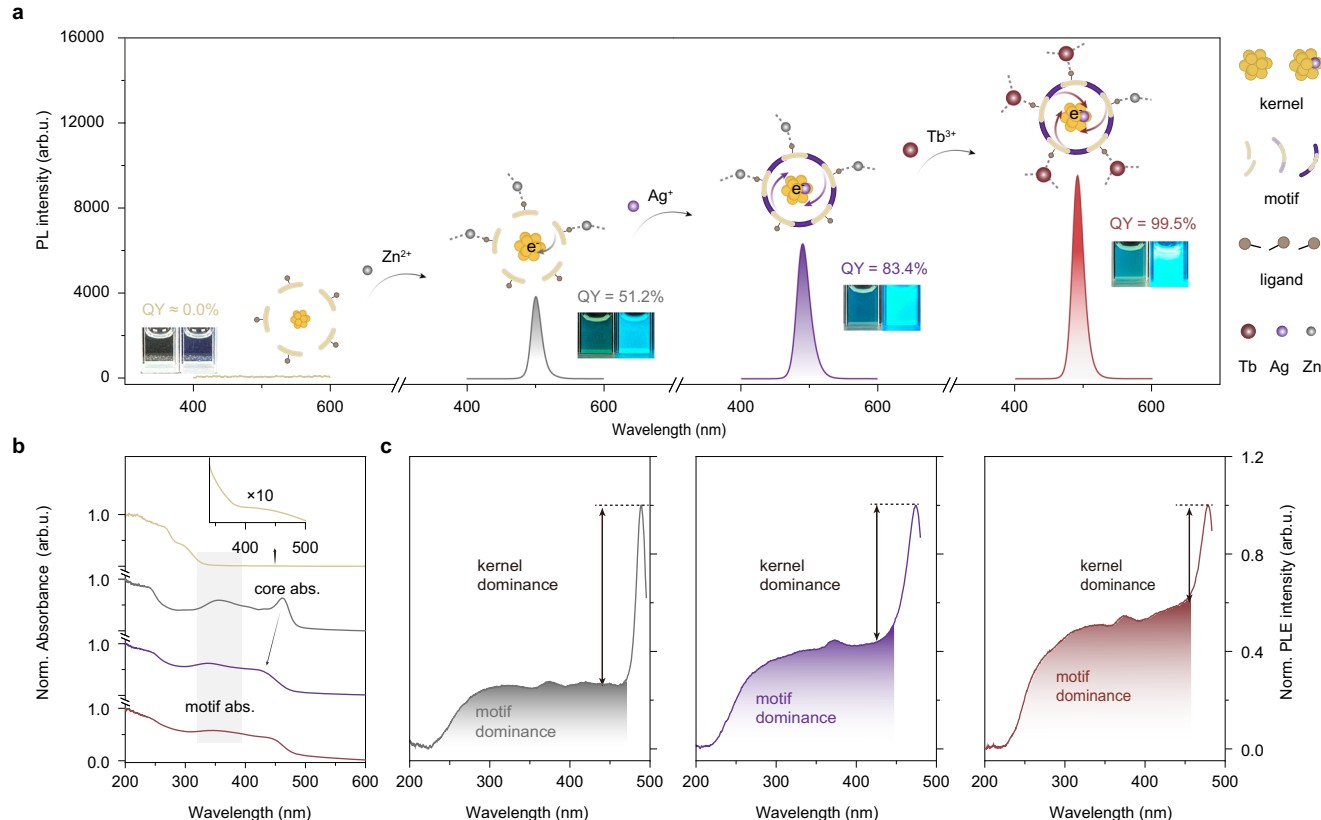

**Fig. 1 | Steady-state optical properties of serial NCs. a** Photoluminescence spectra of **Au, Au-Zn, Au-Zn/Ag,** and **Au-Zn/Ag/Tb** NCs ($\lambda_{ex}$ = 365 nm). Illustrations are photographs taken in visible light (left) and UV light (right). The overall synthetic process of **Au-Zn/Ag/Tb** is shown in the illustrations. **b** Comparison of UV-vis absorption spectra of **Au, Au-Zn, Au-Zn/Ag,** and **Au-Zn/Ag/Tb** NCs. The shaded area represents staple motif absorbance. The yellowish line represents **Au** NCs, the gray line represents **Au-Zn** NCs, the purple line represents **Au-Zn/Ag** NCs and the reddish brown line represents **Au-Zn/Ag/Tb** NCs. **c** Comparison of excitation spectra of **Au-Zn, Au-Zn/Ag,** and **Au-Zn/Ag/Tb** NCs. Source data are provided as a Source Data file.

Da are assigned to the compositions of $[Au_4(MPA)_3 + 2Na\text{-}2H]$, $[Au_4Ag(MPA)_3 + 3Na\text{-}3H]$, respectively. To probe the valence states of the elements and anchoring point of the $Ag^+$ additive, X-ray photoelectron spectra (XPS) and Raman spectra measurements were carried out. The Au $4f_{7/2}$ peak in **Au-Zn** NCs (84.42 eV) is between Au(0) NPs (84.15 eV) and Au(I)-$p$-MBA complex (85.00 eV, Fig. 2a), indicating that **Au-Zn** NCs formed a classical core-shell structure including Au(0) core and Au(I)-S staple motif. After progressively introducing $Ag^+$ and $Tb^{3+}$, the Au $4f_{7/2}$ peaks in **Au-Zn/Ag** (84.54 eV) and **Au-Zn/Ag/Tb** NCs (84.65 eV) exhibit positive shifts of 0.12 and 0.11 eV, respectively. Meanwhile, the Ag $3d_{5/2}$ peaks in **Au-Zn/Ag** (367.89 eV) and **Au-Zn/Ag/Tb** (368.08 eV) NCs are between Ag(0) NPs (367.35 eV) and Ag(I)-$p$-MBA complexes (368.65 eV), suggesting that the existence of Ag(0) and Ag(I) species in the metal core and staple motifs, respectively (Fig. 2b). There are 0.14 eV positive shifts of Ag $3d_{5/2}$ peaks after $Tb^{3+}$ addition. Positive shifts of the binding energy can be rationally assigned to the reduced electron density around Au and Ag, which is determined by many factors, such as the change of the electron distribution of the NCs due to the addition of $Ag^+$, or the coordination interaction between $Tb^{3+}$ and $COO^-$ on the surface of the NCs[39,40]. The doping effect of $Ag^+$ on the NCs is significantly reflected in the different absorption and luminescence behaviors. Compared with **Au-Zn** NCs, the obvious blue shift in the absorption peak of **Au-Zn/Ag** NCs is caused by the strong perturbation to the electronic structure due to the insert of Ag(0) atoms (Supplementary Fig. 6). Generally, Au atoms exhibit stronger $s\text{-}d$ orbital hybridization relative to Ag atoms. After Ag atoms are inserted into the metal core and staple motifs of the NCs, the LUMO (lowest unoccupied molecular orbital) composed of Au $6s$ and $6p$ orbitals greatly changed because of the higher Ag $5s$ orbital than the Au $6p$ orbital[23]. This would

lead to a larger energy gap and, therefore, result in the blue shift of the HOMO (highest occupied molecular orbital)-LUMO electronic transition. Moreover, the $Ag^+$ serves as a linker to bridge the small Au(I)-thiolate motifs on the parental NCs surface to increase the length and rigidity of the staple motifs[41]. In addition, the successful introduction of $Zn^{2+}$ and $Tb^{3+}$ into NCs is evidenced by the XPS spectra (Supplementary Fig. 7).

We then measured Raman spectra to investigate the influence of these heteroatoms on the frequency of structural vibrations (Fig. 2c). With the addition of metal cations, the fingerprint Raman signals less than 200 cm$^{-1}$ (refer to Au-Au bond vibration) and between 200 and 500 cm$^{-1}$ (refer to Au-S bond vibration) both shift to lower-frequency direction, indicating that $Ag^+$ doping in the metal core and staple motifs affects the vibration modes of the Au-Au and Au-S bonds[42]. To demonstrate that the as-prepared NCs are superstructural assemblies, we performed dynamic light scattering (DLS) and transmission electron microscopy (TEM) measurements. Adding $Zn^{2+}$ to Au NCs significantly increases the average size from 2.47 to 230 nm and decreases the zeta potential from − 44.6 to − 36.5 mV (Fig. 2d). With the further addition of $Ag^+$ and $Tb^{3+}$, the size of the resultant NCs increases to 270 and 360 nm and the zeta potential of NCs decreases to − 34.4 and − 30.2 mV (Fig. 2d). TEM gives more evidence of the size growth of NCs, as shown in Supplementary Fig. 8. The small angle X-ray diffraction pattern confirms that the NCs with superstructure are amorphous (Supplementary Fig. 9). The series Au NCs are irregular aggregations. To study the formation and driving forces of the superstructure, $^1$H-nuclear magnetic resonance ($^1$H-NMR) and Fourier transform infrared (FTIR) spectra were carried out. As shown in Fig. 2e and Supplementary Fig. 10, the chemical

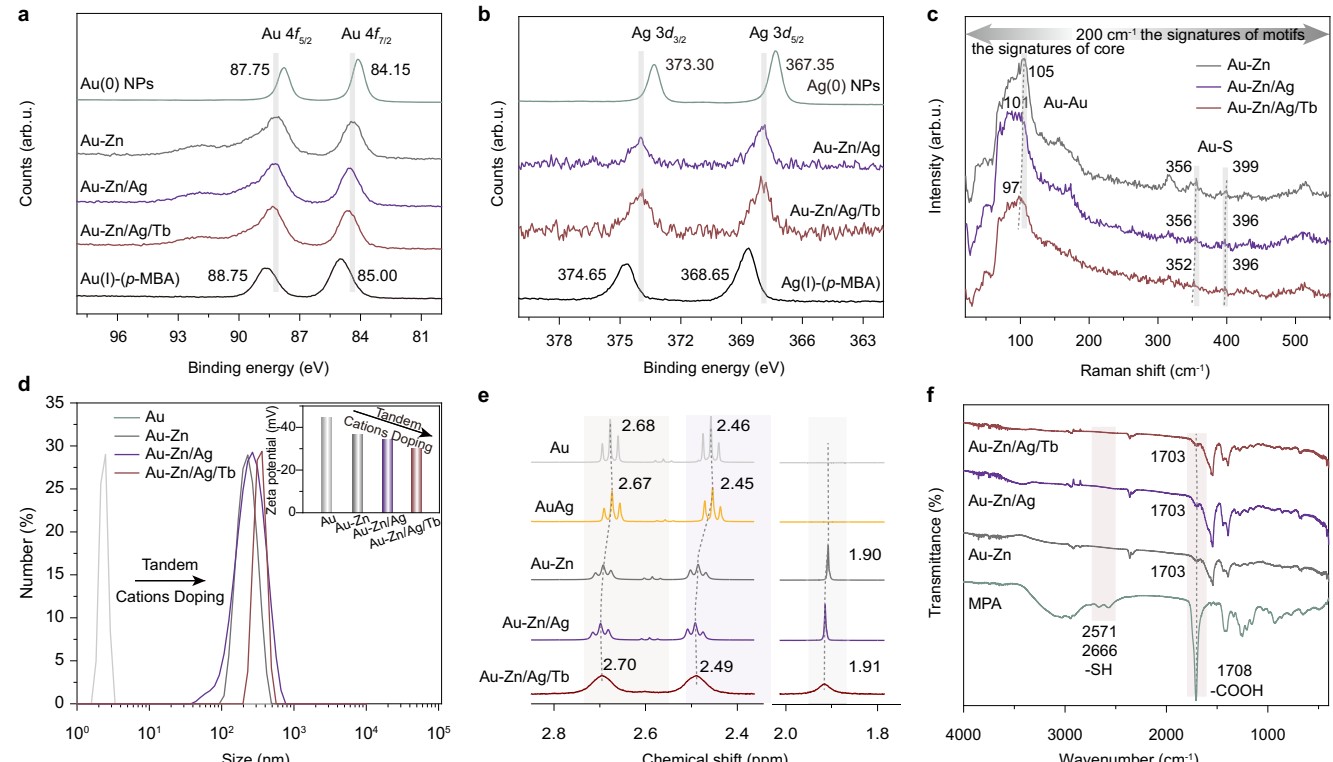

**Fig. 2 | Structural characterizations of serial NCs. a, b** High-resolution Au 4 *f* XPS and Ag 3 *d* XPS spectra of a series of NCs. The gray lines represent the peak center of Au 4 *f* XPS of **Au-Zn** NCs and Ag 3 *d* XPS of **Au-Zn/Ag** NCs in (**a** and **b**), respectively. **c** Raman spectra of **Au-Zn** (gray trace), **Au-Zn/Ag** (purple trace), and **Au-Zn/Ag/Tb** NCs. (reddish-brown trace) NCs. **d** DLS measurement of a series of NCs before and after $Zn^{2+}$, $Ag^{+}$, and $Tb^{3+}$ addition. The inset shows the corresponding zeta potential. **e** $^1$H NMR spectra of serial NCs in $D_2O$. **f** FTIR spectra of MPA ligand, **Au-Zn, Au-Zn/Ag**, and **Au-Zn/Ag/Tb** NCs. Source data are provided as a Source Data file.

shifts of monodisperse MPA ligands are 2.67 and 2.65 ppm. These peaks shift to 2.68 and 2.46 ppm in Au NCs due to the formation of Au-S bonds. After introducing $Zn^{2+}$, $Ag^{+}$, and $Tb^{3+}$ cations, the chemical shifts of MPA ligands further move to the higher field. This result is caused by the decreased electron density around two methylene groups (marked as 1 and 2, Supplementary Fig. 10) on the MPA ligands. However, characteristic MPA peaks in the $^1$H-NMR spectrum of **Au-Zn/Ag/Tb** NCs get broadened, which may be attributed to the shielding effect of the structural vibrations of NCs owing to the strong coordination interaction of $Tb^{3+}$ on the surface of NCs. Meanwhile, metal cations can also induce the formation of hydrogen bonds (1.90 ppm), which is more beneficial to the formation of superstructure. The corresponding FTIR spectra showed that the characteristic carboxylate group peak (at 1708 cm$^{-1}$) on the MPA ligands almost disappeared in three NCs due to the successful coordination of metal cations with them (Fig. 2f)[43,44]. Based on these results, the composition and structural evolution of the prepared samples can be clarified. Firstly, $Zn^{2+}$ ions chelate the $COO^-$ of the surface ligand, which can trigger the appearance of photoluminescence from the **Au-Zn** NCs. Secondly, the introduction of $Ag^+$ ions can simultaneously affect the electronic configuration of the metal core and the rigidity of the interfacial staple motif. Finally, $Tb^{3+}$ ions continue to influence the surface chemistry of the NCs and further reinforce the rigidity of the total NCs.

**Transient-state and temperature-dependent optical properties**
To further explore the effect of sequential cation additives on the optical properties of the NCs, time-resolved PL (TRPL) spectroscopy and temperature-dependent steady-state PL spectroscopy measurements were carried out. All the PL decay curves were recorded by the time-correlated single-photon counting (TCSPC) method and subjected to fitting to extract the PL lifetimes according to the following

monoexponential or double-exponential decay model[27]:

$$I_{(t)} = I_0 + A_1 \exp^{\left(\frac{-t}{\tau_1}\right)} \quad (1)$$

$$I_{(t)} = I_0 + A_1 \exp^{\left(\frac{-t}{\tau_1}\right)} + A_2 \exp^{\left(\frac{-t}{\tau_2}\right)} \quad (2)$$

where $I_{(t)}$ and $I_0$ are the PL intensity measured at time $t$ and $O$. $A_1$ and $A_2$ refer to the constants. $\tau_1$ and $\tau_2$ are the different decay components. According to the double-exponential decay model, the PL average lifetime of **Au-Zn** NCs was fitted to 30.2 ns composed of 18.4 ns non-radiative (42%) and 38.9 ns radiative lifetime (58%) components. The PL average lifetime of **Au-Zn/Ag** NCs increased to 41.1 ns where the non-radiative lifetime increased to 22.3 ns (11%) and radiative lifetime increased to 43.5 ns (89%), respectively. The PL average lifetime of **Au-Zn/Ag/Tb** NCs further increased to 43.6 ns fitted out by the monoexponential decay model (Fig. 3a and Supplementary Table 2). It is observed that the disappearance of non-radiative lifetime was accompanied by the enhancement of PLQY, suggesting that the non-radiative relaxation channels can be greatly suppressed by sequential cationic addition engineering strategy.

The temperature-dependent PL of serial NCs is shown in Fig. 3b and Supplementary Fig. 11. All the emission peaks of serial NCs become intensified with the decline of temperature from 300 to 20 K, indicating the strong electron-vibration interaction. The integrated intensities of emission peaks of **Au-Zn, Au-Zn/Ag**, and **Au-Zn/Ag/Tb** NCs were increased by 2.34, 1.28, and 1.27 times at 20 K, respectively. Therefore, **Au-Zn** NCs are more susceptible to temperatures because many vibrationally related non-radiative relaxation in **Au-Zn** NCs is inhibited at low temperatures. In addition, the red shift of the emission band gets less prominent from **Au-Zn, Au-Zn/Ag**, to **Au-Zn/Ag/Tb**

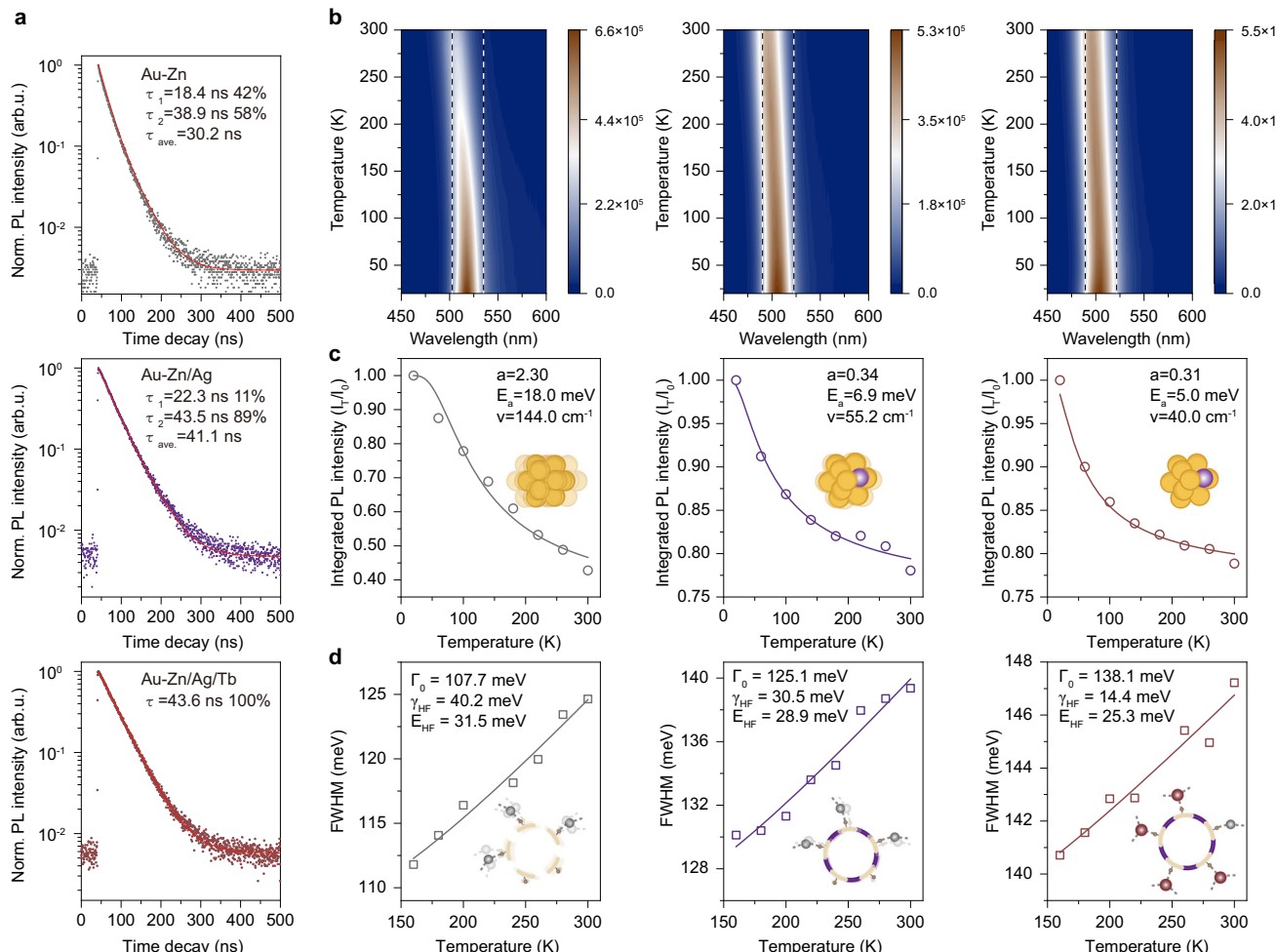

**Fig. 3 | Transient-state and temperature-dependent PL measurements of serial NCs. a** PL decay curve upon 375 nm laser excitation. **b** Temperature-dependent PL map of serial NCs. The excitation source is 365 nm light, and the temperature range is 20–300 K with a temperature interval of 20 K. The scaling of the color scales is the PL intensity. **c** The normalized integrated PL intensity of serial NCs as a function of temperature and their corresponding fitting according to the Arrhenius equation. **d** the PL FWHM of serial NCs as a function of temperature and their corresponding fitting according to the weak electron-vibration coupling approximation model (from left to right: **Au-Zn, Au-Zn/Ag,** and **Au-Zn/Ag/Tb** NCs). Source data are provided as a Source Data file.

NCs at low temperatures, suggesting that the electron-vibration interaction-induced non-radiative relaxations in **Au-Zn** NCs significantly suppressed through sequential addition of $Ag^+$ and $Tb^{3+}$ cation[12,45–47]. The quantitative temperature-dependent PL intensity evolutions for all NCs are plotted as functions of temperature in Fig. 3c. To analyze the thermally activated nonradiative relaxation pathway, the curves of PL intensity versus temperature are fitted according to the following Arrhenius expression[12,48]:

$$I(T) = \frac{I_0}{1 + a e^{-E_a/k_B T}} \quad (3)$$

where $I_0$ and $I_{(T)}$ are the emission intensity at the initial temperature and other testing temperatures, respectively. $a$ is the ratio of non-radiative and radiative probabilities. $E_a$ is the activation energy for the nonradiative relaxation pathway. The fitting results give the activation energies that the low-vibration modes coupled with the emission peaks of the **Au-Zn, Au-Zn/Ag,** and **Au-Zn/Ag/Tb** NCs are 18.0, 6.90, and 5.00 meV, respectively. The corresponding vibration-coupled frequencies are accordingly calculated to be 144.0, 55.2, and 40.0 cm⁻¹, respectively (Fig. 3c). Based on previous theoretical and experimental works[49,50], the possible origin of the relatively low-

frequency vibration mode can be reasonably assigned to the Au-Au bond vibration (typically <200 cm⁻¹). The high-frequency vibration mode should be attributed to the Au-S bond vibration (typically >200 cm⁻¹). In summary, the combination of small Stokes displacement, *ns*-level lifetime, and low-frequency vibration coupling mode of the NCs provide solid proof that the PL of these NCs originates from the metal core. Moreover, the gradual decrease in the vibration-coupled frequency confirms that the vibration of the metal core is significantly suppressed after the sequential introduction of metal ions. In addition, the *a* value also drastically declined from 2.3 for **Au-Zn** NCs to 0.31 for **Au-Zn/Ag/Tb** NCs, which strongly supports the significant suppression of metal core vibration-induced non-radiative decay upon sequential cation additions.

To investigate the additional total structural (surface and interface) vibrations of the NCs, we focus on temperature-dependent emission broadening in >150 K, which is commonly used in semiconductor materials to understand electron-phonon/electron-vibration interactions[51–53]. The temperature-dependent PL FWHM of the three kinds of NCs are further extracted and plotted in Fig. 3d. Interestingly, the temperature-dependent PL FWHM of all NCs in the high-temperature (>150 K) region follows the linear relationship, indicating the common weak electron-vibration coupling approximation mode is

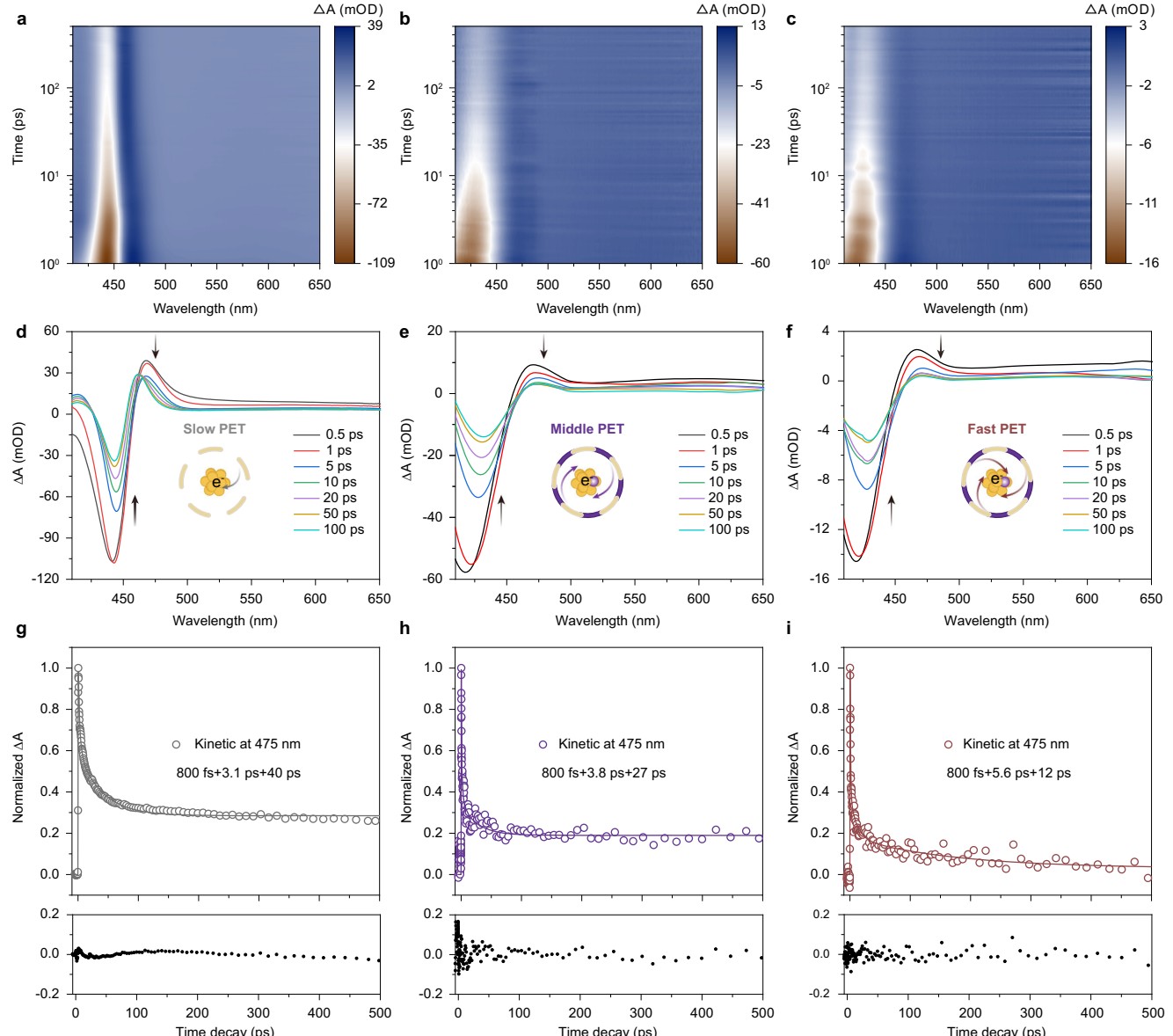

**Fig. 4 | Comparison of fs-TA spectra and their electron dynamics of serial NCs. a–c** The fs-TA map pumped at 365 nm. The scaling of the color scales is the absorption intensity, and the units are milli-optical density. **d–f** The fs-TA profile at different time delays. **g–i** The ESA kinetic decay and fitted residuals within 500 ps (from left to right: **Au-Zn, Au-Zn/Ag,** and **Au-Zn/Ag/Tb** NCs). Source data are provided as a Source Data file.

applicable[28]:

$$\Gamma(T) = \Gamma_0 + \gamma_{LF} T + \gamma_{HF} \frac{1}{\exp\left(\frac{E_{HF}}{k_B T}\right) - 1} \qquad (4)$$

where $\Gamma_0$ is the FWHM of NCs at 0 K, $\gamma_{LF}$ and $\gamma_{HF}$ are the coupling coefficients of an electron with low-frequency vibration (LF) and high-frequency vibration (HF), respectively. $E_{HF}$ is the average energy of the HF vibration. Since we only focus on the coupling of electron-high-frequency vibration caused by staple motif and tail ligand at high temperature, therefore, "$\gamma_{LF} T$" representing the electron-low-frequency vibration is not considered. Accordingly, the weak electron-vibration coupling approximation model is simplified to Eq. (5)[28]:

$$\Gamma(T) = \Gamma_0 + \gamma_{HF} \frac{1}{\exp\left(\frac{E_{HF}}{k_B T}\right) - 1} \qquad (5)$$

It is worth noting that the coupling coefficients of the electron with high-frequency vibrations are significantly reduced with sequential cation addition ($\gamma_{HF} = 40.2$, 30.5, and 14.4 meV for **Au-Zn, Au-Zn/Ag,** and **Au-Zn/Ag/Tb**, respectively). Therefore, the addition of metal cations can also able to suppress the coupling of excited electrons and high-frequency vibration in the surface and interface in metal NCs (Fig. 3d).

## Ultrafast excited-state dynamics

The core-state PL of all NCs can still be detected when we use 365 nm light to excite the staple motif (Fig. 1). Since the low- and high-energy excitations mainly contributed from the Au(0) core and surface Au(I)-S components, respectively, we speculate that there is a PET channel to transfer excited electrons from the interfacial staple motif to the metal core. To prove this, femtosecond-transient absorption (fs-TA) measurements of serial NCs were performed (Fig. 4a–f and Supplementary Fig. 12). As shown in Fig. 4a–f, upon photo-excitation with a 365 nm laser pulse, an apparent ground-state bleaching (GSB) can be identified

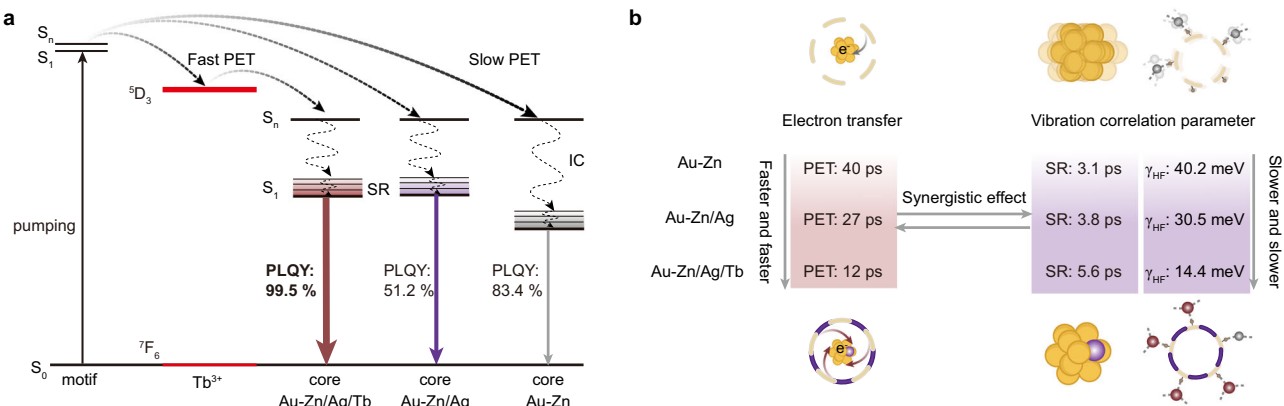

**Fig. 5 | Schematic illustration of the excited-state electron dynamics of serial NCs. a** Schematic diagram illustrating the excited-state dynamics of serial NCs. Arrows denote in (**a**) the transitions between different electronic states. **b** Changes in the time constant of PET, SR, and coupling strength of electrons-high-frequency vibration from NCs surface and interface. $S_0$: ground state, $S_n$: high singlet state, $S_1$: lowest singlet state, IC: internal conversion; SR: structural relaxation; PET: photoinduced electron transfer.

at 443 nm in the TA spectra of **Au-Zn** NCs. After incorporating $Ag^+$ and $Tb^{3+}$, their GSB peaks position blue-shifts to 432 and 429 nm for **Au-Zn/Ag** and **Au-Zn/Ag/Tb** NCs, respectively. Following their GSB band, a broad excited-state absorption (ESA) band contributing by staple motif is observed, manifesting the dense excited states of serial NCs[54]. Notably, no typical SE signals were acquired, of which similar scenarios were also documented, even for the metal NCs with high PLQY[55,56], which may be caused by their blending with stronger GSB signals and/or the competition between SE and ESA processes[57,58]. The ESA decays were subjected to fitting to extract different relaxation processes of the excited-state electrons in serial NCs. Within an approximate 500 ps time window, three decay components were fitted out in the ESA kinetics: 800 fs, 3.1 ps, and 40 ps for **Au-Zn** NCs; 800 fs, 3.8 ps, and 27 ps for **Au-Zn/Ag** NCs; 800 fs, 5.6 ps, and 12 ps for **Au-Zn/Ag/Tb** NCs (Fig. 4g–i and Supplementary Table 3). Notably, the excitation energy of 3.40 eV (365 nm) is larger than HOMO-LUMO gap energies and can pump the ground state electrons to the hot $S_n$ state (Supplementary Fig. 4). Therefore, this ultrafast decay process (800 fs) should be better attributed to the internal conversion (IC) process of hot electrons from $S_n$ to the $S_1$ state ($S_n \rightarrow S_1$). The several picosecond time components (3.1, 3.8, and 5.6 ps) are assigned to the core-directed structural vibration in serial NCs[49]. Finally, the relatively slow decay process (40, 27, and 12 ps) can be relevant to the PET process from the excited state of staple motifs to the metal core[15,43,59]. It can be seen that the time component of core-directed structural vibration gradually increases, indicating that the vibration of the metal core is greatly suppressed after cation additions. Intriguingly, the time component of dozens of picoseconds gradually decreases. This can be explained by the fact that the excited state electrons from the staple motif transfer to the metal core faster. The relaxation processes of excited electrons mentioned above are mainly related to the composition and compactness of metal NCs and abundant energy levels of rare earth ions.

Combined with the measurements of steady absorption, PL spectra, and ultrafast excited-state dynamics, we are ready to give a rational explanation for the observed near-unity PLQY of NCs induced by sequential cations addition engineering (Fig. 5). First, the electrons in the interfacial staple motifs can be excited from the ground state ($S_0$) to the excited state ($S_n$) by the high-energy (3.40 eV) pump. Then, the excited state electrons in staple motifs spontaneously transfer into the $S_n$ excited state of the metal core. Subsequently, they experience a fast decay into the $S_1$ state of the metal core through the IC process ($S_n \rightarrow S_1$). After that, the excited electrons in the $S_1$ state non-radiatively relax into the lowest energy level of $S_1$ through structural relaxation. Finally, the radiative relaxation of excited-state electrons from the $S_1$ to

the $S_0$ ground state in the metal core can emit ~ 490 nm fluorescence with a *ns*-level lifetime.

The introduction of metal cations has two main effects on the PLQY enhancement of NCs: (i) The PET process is more efficient, and (ii) the introduction of $Ag^+$ will decrease the distance from the staple motif to the metal core and, therefore, increase the rigidity of the structure. The efficiency of PET increases exponentially with decreasing distance (*d*) as indicated by Eq. (6), where $\beta$ is the PET distance decay constant[58,60]. Thus, this provides solid evidence that $Ag^+$ accession facilitates PET from the staple motif to the metal core.

$$E_{ET} \propto \exp(-\beta d) \qquad (6)$$

More interestingly, rare earth ions have abundant energy levels[61,62]. The $^5D_3$ level of $Tb^{3+}$ is located between the $S_1$ level of the staple motif and the $S_1$ level of the metal core, which serves as a bridge to promote the PET process. Moreover, introducing rare earth cations can further coordinate the uncoordinated carboxylate group at the surface of the NCs. This configuration further increases the rigidity of the structure and reduces the total-structure vibration-related non-radiative process. To sum up, the $Ag^+$ addition in the metal core expands the energy gap. While $Ag^+$ addition in the staple motif can increase the rigidity of the structure and suppress the non-radiative relaxation caused by the vibration of the metal core and staple motif. The $Tb^{3+}$ addition can further coordinate the naked carboxylate group at the surface of NCs, further increasing the rigidity of the structure and reducing the non-radiative relaxation caused by the vibration of the outermost ligands. As a result, the near-unity PLQY NCs can be achieved due to the synergistic effect of the suppression of total-structural vibrations and the efficient PET process through a sequential metal cation engineering strategy.

### The universality of the proposed strategy
To verify the universality of the proposed sequential metal cation engineering strategy and prove that the energy level of rare earth ions plays a key role in PET, we added different rare earth ions ($Ce^{3+}$, $Pr^{3+}$, $Nd^{3+}$, $Sm^{3+}$, $Eu^{3+}$, $Dy^{3+}$, $Ho^{3+}$, $Er^{3+}$, $Tm^{3+}$, and $Yb^{3+}$) into the **Au-Zn/Ag** NCs (Supplementary Fig. 13) and characterized their steady-state and fs-TA spectra. The emission peak positions of other rare earth-contained NCs are hardly changed (Fig. 6a). The time constants related to PET from the staple motif to the metal core changed with the addition of different rare earth ions (Fig. 6b, Supplementary Fig. 14 and Table 1). According to the doping effect, we divided the rare earth ions into two categories. The time constants of PET in $Ce^{3+}$, $Gd^{3+}$, and $Yb^{3+}$-contained

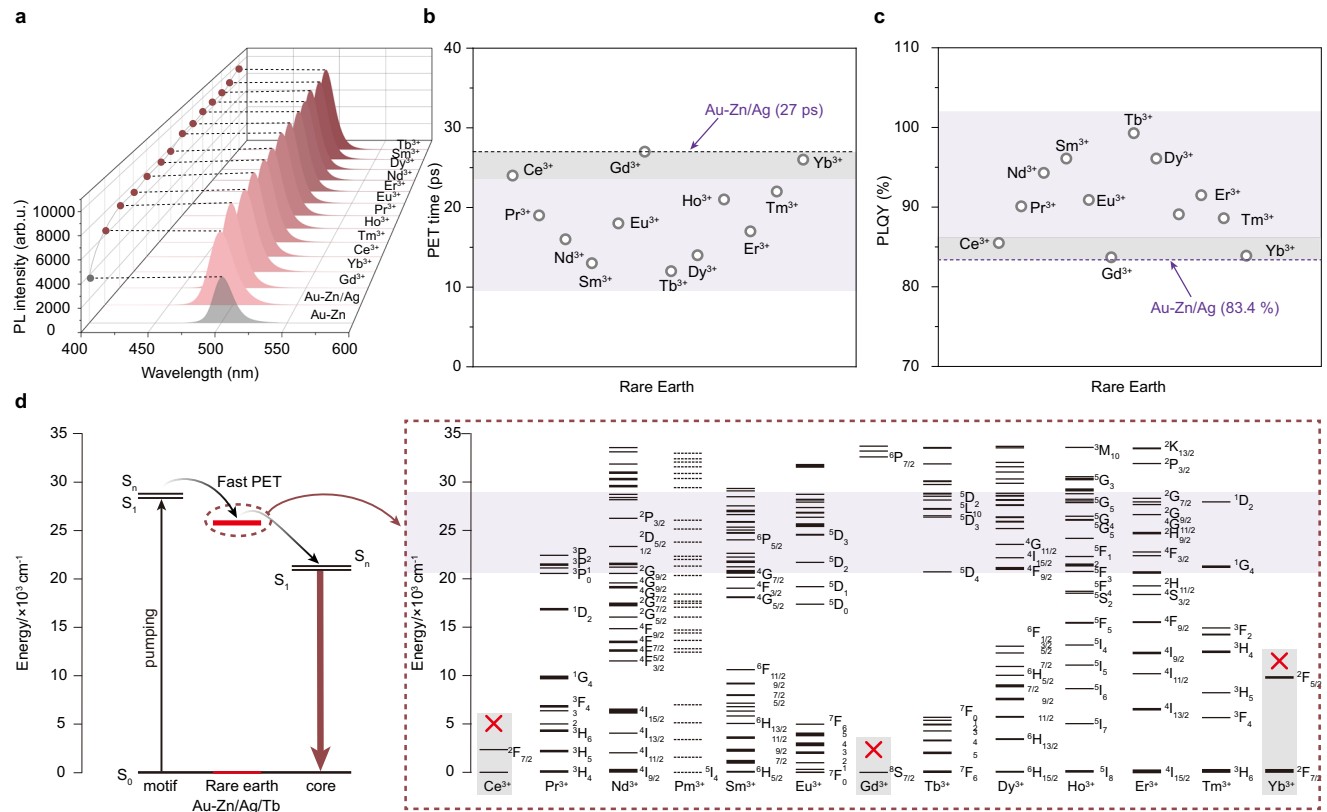

**Fig. 6 | The universality of different rare earth ions. a** Photoluminescence spectra of **Au-Zn/Ag/R** NCs ("R" denotes arbitrary rare earth ions). **b** The comparison of PET between **Au-Zn/Ag** and **Au-Zn/Ag/R** NCs. The gray shaded area represents a similar PET time for **Au-Zn/Ag/R** and **Au-Zn/Ag**. The reddish-brown shaded area represents different degrees of shortening of the **Au-Zn/Ag/R** PET time. **c** The comparison in PLQY between **Au-Zn/Ag** and **Au-Zn/Ag/R** NCs. The gray shaded area represents a similar PLQY for **Au-Zn/Ag/R** and **Au-Zn/Ag**. The reddish-brown shaded area represents different degrees of increase of the **Au-Zn/Ag/R** PLQY. **d** Schematic illustration of rare earth ions acting as an electron transfer bridge and energy level diagram of rare earth ions. The reddish-brown shaded area indicates that the rare earth can act as an electron transfer bridge, while the rare earth ions in the gray-shaded area with a cross sign cannot. Source data are provided as a Source Data file.

NCs (the gray region) are close to that in **Au-Zn/Ag** NCs. However, the time constant of other rare earth ions-contained NCs decreases to different degrees. Their PLQY and the time constant associated with PET show the opposite trend (Fig. 6c). This result is attributed to the discrepancy in the energy level position of different rare earth ions. As shown in Fig. 6d, the ideal rare earth ions to act as a PET bridge are the ones whose energy level should lie between the $S_1$ level of the staple motif and the $S_1$ level of the metal core (the light purple region).

## Table 1 | Fitting parameters of normalized TA of Au-Zn/Ag/ R NCs

| Rare earth | $A_1 (\times 10^{-3})$ | $\tau_1$ (fs) | $A_2 (\times 10^{-3})$ | $\tau_2$ (ps) | $A_3 (\times 10^{-3})$ | $\tau_3$ (ps) |
|---|---|---|---|---|---|---|
| $Tb^{3+}$ | 2.441 | 800 | 0.141 | 5.6 | 0.454 | 12 |
| $Sm^{3+}$ | 2.417 | 800 | 0.149 | 5.6 | 0.741 | 13 |
| $Dy^{3+}$ | 2.881 | 800 | 0.115 | 5.6 | 0.984 | 14 |
| $Nd^{3+}$ | 2.670 | 799 | 0.230 | 5.2 | 0.983 | 16 |
| $Er^{3+}$ | 2.216 | 800 | 0.128 | 5.0 | 0.595 | 17 |
| $Eu^{3+}$ | 2.637 | 801 | 0.116 | 4.8 | 0.274 | 18 |
| $Pr^{3+}$ | 2.915 | 800 | 0.115 | 4.7 | 0.938 | 19 |
| $Ho^{3+}$ | 2.432 | 801 | 0.197 | 4.3 | 0.349 | 21 |
| $Tm^{3+}$ | 2.911 | 800 | 0.460 | 4.2 | 0.167 | 22 |
| $Ce^{3+}$ | 2.666 | 800 | 0.720 | 4.0 | 0.207 | 24 |
| $Yb^{3+}$ | 3.253 | 799 | 0.183 | 3.8 | 0.222 | 26 |
| $Gd^{3+}$ | 2.515 | 800 | 0.322 | 3.8 | 0.317 | 27 |

Keeping this principle in mind, we found rare earth ions like $Tb^{3+}$, $Sm^{3+}$, and $Dy^{3+}$ can effectively promote the PET process. However, the intrinsic non-radiative relaxation caused by cross-relaxation between multiple energy levels of rare earth ions may interfere with the effect of PET. For example, the PET of $Ho^{3+}$ and $Er^{3+}$ is not very efficient, and the PLQY is not near-unity even though they have suitable energy levels to act as bridges. Moreover, rare earth ions such as $Ce^{3+}$, $Gd^{3+}$, and $Yb^{3+}$ with very low energy levels cannot efficiently construct PET bridges. Therefore, the PLQY of $Ce^{3+}$, $Gd^{3+}$, and $Yb^{3+}$ rare earth-chelated NCs is close to pristine **Au-Zn/Ag** NCs. In addition, **Au-Zn/Cu/Tb** NCs were synthesized based on similar methods, which present a similar enhancement effect (Supplementary Fig. 15). After the introduction of $Cu^{2+}$ and $Tb^{3+}$ into **Au-Zn** NCs, the contribution of the PET band to luminescence gradually increases in their excitation spectra, indicating that the PET from the staple motif to the metal core gets more efficient. As shown in Supplementary Fig. 16, the introduction of $Cu^{2+}$ can increase the energy gap of the NCs, which is similar to the results of **Au-Zn/Ag/Tb** NCs. We found **Au-Zn/Cu/Tb** NCs also achieve near-unity PLQY (96.4%) due to the similar working mechanism of total-structure vibration suppression and excellent PET process (Supplementary Fig. 18a). In addition, we also investigated the effects of other rare earth ions on the photoluminescence of **Au-Zn/Cu** NCs (Supplementary Fig. 17). As expected, rare earth ions with lower energy levels (such as $Ce^{3+}$, $Gd^{3+}$, and $Yb^{3+}$) also have lower PLQY close to **Au-Zn/Cu** NCs, while the PLQY of NCs added with other rare earth ions increased with different degrees (Supplementary Fig. 18b). These results demonstrate

the universality of the as-developed sequential cations addition strategy for luminescence enhancement of metal NCs.

## Discussion

In conclusion, through sequential cations addition engineering, we achieved near-unity PLQY of 99.5% in aqueous Au NCs at room temperature. The sequential decoration of Au NCs with $Zn^{2+}$, $Ag^+$, and $Tb^{3+}$ cantions can significantly suppress the total structural motions and modulate the electron transfer dynamics. Especially, the alloy of $Ag^+$ in the metal core greatly disturbs the electronic structure of the metal core and reduces the vibration. The alloy of $Ag^+$ in the staple motifs can rigidify the interfacial structure, giving rise to the accelerated electron transfer from staple motifs to the metal core. The coordination of $Tb^{3+}$ on the deprotonated carboxyl groups in Au NCs further suppresses the vibration of surface ligands. More importantly, the introduced $Tb^{3+}$ provides an additional ladder-like energy level to boost the electron transfer. The universality of the intermediate bridge was also reproduced by using other rare earth ions ($Sm^{3+}$, $Nd^{3+}$, $Dy^{3+}$) with suitable energy level positions. In addition, substituting $Ag^+$ with $Cu^{2+}$ to doping into the Au NCs can also enable 96.4% PLQY. This study provides an in-depth understanding of the structure-luminescence relationship and luminescence mechanism of metal NCs superstructures, which provides a promising approach for the rational and delicate design of high-efficiency metal NCs, and hopefully, it can be used in medical diagnosis and, optical sensing, biomedical imaging, and luminescent displays.

## Methods

### Materials and reagents

Hydrogen tetrachloroaurate(III) trihydrate ($HAuCl_4 \cdot 3H_2O$, ≥49.0 Au basis) was purchased from Sigma Aldrich. Silver nitrate ($AgNO_3$, 99.99% Ag basis), Cupric chloride ($CuCl_2$, 99.99%), 3-Mercaptopropionic acid (MPA, 99%), Sodium hydroxide (NaOH, 96%), Hydrochloric acid (12 M), Zinc acetate (99.995% metals basis), Cerium(III) nitrate hexahydrate (99.99%), Praseodymium(III) chloride hexahydrate (99.99%), Neodymium nitrate hexahydrate (99.99%), Samarium(III) chloride (98%), Europium(III) nitrate hydrate (99.99%), terbium nitrate hexahydrate (99.99%), Dysprosium nitrate hexahydrate (99.99%), Holmium(III) chloride (99.99%), Erbium(III) nitrate hexahydrate (99.99%), Thulium nitrate hexahydrate (9.99%), Yttrium(III) nitrate tetrahydrate (99.99%) were purchased from Aladdin Reagent Co. (Shanghai, China); All chemicals were used as received without additional purification.

### Synthesis of Au NCs

In a typical synthesis of isolated Au NCs, 400 μL aqueous solution of $HAuCl_4$ (50 mM) was added to 4.6 mL of ultrapure water under 600 rpm stirring at room temperature. Then, 69 μL of MPA was added to the above aqueous solution. The light-yellow solution (the original color of the $HAuCl_4$ solution) quickly turned white with the appearance of precipitation, implying the formation of Au(I)-(MPA) complexes. After stirring for 15 min, a certain amount of NaOH (1 M) was dropped into the reaction solution to tune the pH of the aqueous solution to 7.90 to dissolve the white precipitation. After incubating for 24 h, isolated Au NCs with non-photoluminescence can be required.

### Synthesis of AuAg NCs

In a typical synthesis of isolated AuAg NCs, 320 μL aqueous solution of $HAuCl_4$ (50 mM) and 80 μL aqueous solution of $AgNO_3$ (50 mM) was added to 4.6 mL of ultrapure water under 600 rpm stirring at room temperature. Then, 69 μL of MPA was added to the above aqueous solution. The light-yellow solution (the original color of the $HAuCl_4$ solution) quickly turned white with the appearance of precipitation, implying the formation of Au(I)@Ag(I)-(MPA) complexes. After stirring for 15 min, a certain amount of NaOH (1 M) was dropped into the

reaction solution to tune the pH of the aqueous solution to 7.90 to dissolve the white precipitation. After incubating for 24 h, isolated AuAg NCs with non-photoluminescence can be required.

### Synthesis of Au-Zn NCs

In a typical synthesis of **Au-Zn** NCs, 400 μL aqueous solution of $HAuCl_4$ (50 mM) was added to 4.6 mL of ultrapure water under 600 rpm stirring at room temperature. Then, 69 μL of MPA was added to the above aqueous solution. The light-yellow solution (the original color of the $HAuCl_4$ solution) quickly turned white with the appearance of precipitation, implying the formation of Au(I)-(MPA) complexes. After stirring for 15 min, a certain amount of NaOH (1 M) was dropped into the reaction solution to tune the pH of the aqueous solution to 7.90 to dissolve the white precipitation. To trigger the assembly of isolated Au NCs, a 1 mL aqueous solution of Zn(OAc)₂ (0.1 M) was added. After incubating for 24 h, **Au-Zn** NCs with bright blue emission can be required.

### Synthesis of Au-Zn/Ag NCs

In a typical synthesis of **Au-Zn/Ag** NCs, 320 μL aqueous solution of $HAuCl_4$ (50 mM) and 80 μL aqueous solution of $AgNO_3$ (50 mM) was added to 4.6 mL of ultrapure water under 600 rpm stirring at room temperature. Then, 69 μL of MPA was added to the above aqueous solution. The light-yellow solution (the original color of the $HAuCl_4$ solution) quickly turned white with the appearance of precipitation, implying the formation of Au(I)@Ag(I)-(MPA) complexes. After stirring for 15 min, a certain amount of NaOH (1 M) was dropped into the reaction solution to tune the pH of the aqueous solution to 7.90 to dissolve the white precipitation. To trigger the assembly of isolated AuAg NCs, a 1 mL aqueous solution of Zn(OAc)₂ (0.1 M) was added. After incubating for 24 h, **Au-Zn/Ag** NCs with bright blue emission can be required. In addition, **Au-Zn/Ag** NCs with different Au and Ag content were synthesized using the same method.

### Synthesis of Au-Zn/Ag/Tb NCs

In a typical synthesis of **Au-Zn/Ag/Tb** NCs, 320 μL aqueous solution of $HAuCl_4$ (50 mM) and 80 μL aqueous solution of $AgNO_3$ (50 mM) was added to 4.6 mL of ultrapure water under 600 rpm stirring at room temperature. Then, 69 μL of MPA was added to the above aqueous solution. The light-yellow solution (the original color of the $HAuCl_4$ solution) quickly turned white with the appearance of precipitation, implying the formation of Au(I)@Ag(I)-(MPA) complexes. After stirring for 15 min, a certain amount of NaOH (1 M) was dropped into the reaction solution to tune the pH of the aqueous solution to 7.90 to dissolve the white precipitation. Then, 100 μL of rare earth salt solution (50 mM) was added to the above aqueous solution. Finally, a 1 mL aqueous solution of Zn(OAc)₂ (0.1 M) was added. After incubating for 24 h, **Au-Zn/Ag/Tb** NCs with bright blue emission can be required. In addition, we also adjusted the content of rare earth ions (50 mM, 0 μL, 50 μL, 100 μL, and 150 μL) and used other rare earth ions ($Ce^{3+}$, $Pr^{3+}$, $Nd^{3+}$, $Sm^{3+}$, $Eu^{3+}$, $Dy^{3+}$, $Ho^{3+}$, $Er^{3+}$, $Tm^{3+,}$ and $Yb^{3+}$) to synthesize series of NCs.

### Synthesis of Au-Zn/Cu NCs

In a typical synthesis of **Au-Zn/Cu** NCs, 320 μL aqueous solution of $HAuCl_4$ (50 mM) and 80 μL aqueous solution of $CuCl_2$ (50 mM) was added to 4.6 mL of ultrapure water under 600 rpm stirring at room temperature. Then, 69 μL of MPA was added to the above aqueous solution. The light-yellow solution (the original color of the $HAuCl_4$ solution) quickly turned brown with the appearance of precipitation, implying the formation of Au(I)@Cu(II)-(MPA) complexes. After stirring for 15 min, a certain amount of NaOH (1 M) was dropped into the reaction solution to tune the pH of the aqueous solution to 7.90 to dissolve the brown precipitation. To trigger the assembly of isolated AuCu NCs, a 1 mL aqueous solution of Zn(OAc)₂ (0.1 M) was added.

After incubating for 24 h, **Au-Zn/Cu** NCs with bright blue emission can be required.

## Synthesis of Au-Zn/Cu/Tb NCs

In a typical synthesis of **Au-Zn/Cu/Tb** NCs, 320 µL aqueous solution of $HAuCl_4$ (50 mM) and 80 µL aqueous solution of $CuCl_2$ (50 mM) was added to 4.6 mL of ultrapure water under 600 rpm stirring at room temperature. Then, 69 µL of MPA was added to the above aqueous solution. The light-yellow solution (the original color of the $HAuCl_4$ solution) quickly turned brown with the appearance of precipitation, implying the formation of Au(I)@Cu(II)-(MPA) complexes. After stirring for 15 min, a certain amount of NaOH (1 M) was dropped into the reaction solution to tune the pH of the aqueous solution to 7.90 to dissolve the brown precipitation. Then, 100 µL of rare earth salt solution (50 mM) was added to the above aqueous solution. To trigger the assembly of isolated AuCu NCs, a 1 mL aqueous solution of $Zn(OAc)_2$ (0.1 M) was added. After incubating for 24 h, **Au-Zn/Cu/Tb** NCs with bright blue emission can be required. In addition, we also used other rare earth ions ($Ce^{3+}$, $Pr^{3+}$, $Nd^{3+}$, $Sm^{3+}$, $Eu^{3+}$, $Dy^{3+}$, $Ho^{3+}$, $Er^{3+}$, $Tm^{3+}$, and $Yb^{3+}$) to synthesize a series of NCs.

## Characterization

Matrix-assisted laser desorption ionization-time-of-flight (MALDI-TOF) mass spectra were tested on Brucker Autoflex speed TOF under a positive linear mode. Trans-2-[3-(4-tert-Butylphenyl)-2-methyl-2-propenylidene] malononitrile (DCTB) was employed as the matrix for all samples. Specific details are described as follows: The 10 µL DCTB matrix and 10 µL NCs aqueous solution were fully mixed. Then, spot 1.5 µL of the mixed solution on the target plate and wait for the solvent to evaporate. The detector gain was set at 10. The spot size was fixed at medium, and the laser intensity was tuned to be greater than 70%. X-ray photoelectron spectra (XPS) of the serials NCs superstructures were carried out on an ESCALAB250 spectrometer (Thermo Fisher). The diffraction of X-rays (XRD) was conducted by using the grazing incidence (GI) mode in Rigaku Smartlab 9 kW. Transmission electron microscopy (TEM) images were performed on the FEI Tecnai G2 F20 microscopy operated at 200 kV. FTIR spectra were collected on a Nicolet Is50 spectrometer (Thermo Fisher) using a single attenuated total reflectance accessory at 400–4000 $cm^{-1}$ scanning range. Raman spectra were performed on a high-resolution laser Raman spectrometer (HORIBA Jobin Yvon) attached with a 2017 Argon ion gas laser with a 633 nm laser excitation source and the scanning range of Raman spectra were 40–500 $cm^{-1}$. $^1$H-NMR measurements were recorded with AS 400 MHz (Q. One Instruments Ltd.) system using $D_2O$ as a solvent for all $^1$H-NMR measurements. UV-vis absorption spectra were recorded on the Shimadzu UV-1900i spectrometer. PL and PL excitation (PLE) spectra were captured on a Hitachi F-4700 spectrometer, and the excitation wavelength of all the tests mentioned in this paper is selected to be 365 nm. The PLQY of all samples in aqueous solution was measured and calculated on a QM8000 HORIBA spectrometer attached with an integrating sphere coating with a reflective $BaSO_4$ layer. For the detailed measurements of the PLQY of all samples, an Xe lamp with a fixed emission wavelength at 365 nm was used as the excitation source. Pure water support was measured first in the integrating sphere and used as blank references for the samples in an aqueous solution. The calculation of absolute PLQY values was conducted on the built-in "FelixGX" software (version 4.9.0.10321). The values of PLQY were calculated by using the following equation: $PLQY = \frac{\int \lambda P(\lambda)d\lambda}{\int \lambda \{E(\lambda) - R(\lambda)\}d\lambda}$, where E(λ)/hv, R(λ)/hv, and P(λ)/hv is the number of photons in the spectrum of excitation, reflectance, and emission, respectively. Photoluminescence lifetime measurements were performed through a TCSPC method on an Edinburgh FLS1000 spectrofluorometer with pulsed laser excitation sources.

Temperature-dependent PL spectra of NCs film were measured on FLS1000 with an Oxford Optistat attemperator cooled by liquid nitrogen. Femtosecond-TA spectroscopy was conducted under a pump wavelength of 365 nm on HELIOS (Ultrafast systems) spectrometers.

## Reporting summary

Further information on research design is available in the Nature Portfolio Reporting Summary linked to this article.

## Data availability

The data supporting the findings of this study are available within the paper, Supplementary Information, and Source Data files. Extra data are available from the author upon request. Source data are provided in this paper.

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

## Acknowledgements

This work was supported by the National Natural Science Foundation of China (NSFC) (T2325015 and U21A2068 to X.B., 61935009 to Y. Zha. 12174151 to Z.W., 12304448 to T.L.).

## Author contributions

X.W. and Y.Zho. contributed equally to this work. X.W., Z.W., and A.T. conceptualized the idea and co-supervised this work. Y.Zho. carried out the serial femtosecond-TA experiments. X.B., X.W., T.L., M.L., and Y.Zha. performed the synthesis of metal NCs and optical spectra measurements. X.W., Y.Zho., K.W., W.D., M.L., X.B., Z.W., and Y.Zha. discussed the experimental data and commented on the original draft. X.W., Y.Zho., and Z.W. wrote the manuscript.

## Competing interests

The authors declare no competing interest.
