## [Transparent Peer Review file · Nature Communications]

Sequential addition of cations increases photoluminescence quantum yield of metal nanoclusters near unity

Corresponding Author: Professor Zhennan Wu

Version 0:

Reviewer comments:

Reviewer #1

(Remarks to the Author)

Luminescent metal nanoclusters with ultra-high photoluminescence quantum efficiency (PLQY) and narrow emission bands are long-pursued for researchers but remain a challenge. This work is very impressive in that the authors report the near-unity PLQY of Au nanoclusters with narrow full width at half maximum (FWHM) of 17-22 nm by the as-proposed sequential cation additives strategy. The progressive introduction of Zn²⁺, Ag⁺, and Tb³⁺ cations strategy is novel and the claimed mechanism of efficient suppression of the total structural motions and management of the electron transfer dynamics is particularly interesting. In addition, the author validated the universality of their strategy, by employing different coin metal architectures and other lanthanide ions. More importantly, the lanthanide ions provide a hopping platform for the excited electrons as their intrinsic ladder-like energy level structure. Overall, this manuscript is well-structured, and the conclusions are well-supported by the presented data. This manuscript meets the high standards of Nature Communications and deserves to be published. Therefore, I strongly recommend accepting this manuscript after solving the following minor issues:

1. The Zn²⁺ can trigger the enhanced emission of the optical-silenced Au NCs by chelating the COO⁻ of the surface ligand. In addition to Zn²⁺ cations, what are the effects of other cations, such as Ca²⁺ or Mg²⁺, on the photoluminescence boost of the NCs?
2. In Fig. 4, I suggest giving an error bar or error region, and necessary residual and χ^2 in the fitting of the ESA kinetics to judge the fitting quality. In addition, I think the curve's color is too similar in Fig. 4d-f to be identified. I suggest that the authors use some more easily distinguished colors.
3. The authors claimed that "...which is commonly used in semiconductor materials to understand electron-phonon interactions.", please give the corresponding references.
4. Before publication, details regarding the experimental conditions for MALDI-MS, such as the matrix, the ratio of Au NCs to matrix, and the instrument, should be included in the Supporting Information.
5. In addition to the chelation of Tb³⁺ with the surface ligands of NCs, the Tb³⁺ also provides a hopping platform for the excited electrons from the staple motif to the metal core. How can the authors prove that the energy level of Tb³⁺ is located between the Sn of the staple motif and the S₂/S₁ energy levels of the metal core?

Reviewer #2

(Remarks to the Author)

In this manuscript, Wang et al. synthesized 3-mercaptopropionic acid (MPA) protected Au nanoclusters and introduced Zn²⁺, Ag⁺ and rare earth Tb³⁺ ions to improve the PL intensity of the as-synthesized nanoclusters. MALDI-MS, XPS and DLS were employed to understand the composition of the nanoclusters. Low-temperature PL analysis and TAS were also used to study the electron dynamic and PL mechanism.

This work is interesting and suitable for publication in Nat Commun after addressing the following questions.

1. Line 173, it would be better to use meV to describe FWHM. Because, on the energy scale, 22 nm linewidth in the blue region is much larger than the same 22 nm linewidth in the NIR region.
2. In this work, the analysis of the composition of the nanoclusters heavily relies on the MALDI-MS results. In fig. s6a, there is a strong peak on the left side of the 1147.3 peak and a weak peak on the right side of the 1147.3 peak, please assign them.
3. Line 213, the author mentioned the addition of Tb³⁺ affects the electron density of silver. Why are Au atoms not affected? If the mechanism is correct, Au should also be affected because it's the majority element in the cluster.
4. Line 223, the author mentioned "the added Ag⁺ serves as a linker to bridge the small Au(I)-thiolate motifs on the parental NCs surface to increase the length and rigidity of the staple motifs". So, how does Ag⁺ serve as the linker? And, why is Ag⁺ itself not enough ("isolated AuAg NCs with non-photoluminescence") to make the MPA-protected nanocluster emissive that Zn must be introduced?
5. Line 235, the author claimed "Adding Zn²⁺ to Au NCs significantly increases the average size from 2.47 to 270 nm". Is it possible to observe such a size increase under TEM?
6. Were the "300 to 20 K" measurements done on solutions or films? No experimental detail can be found.
7. Line 406, the author mentioned "the excited state electrons in staple motifs spontaneously transfer into the S2 excited state of the metal core due to the minimum energy discrepancy". Is there any evidence that supports this energy argument or is it just a guess? Similarly, line 421, "The 5D3 level of Tb³⁺ is located between the Sn level of the staple motif and the S2 level of the metal core".
8. The author claimed electron transfer from the nanocluster to Tb³⁺ and back to the nanocluster, which is not convincing. In Fig. 5, the author claimed PET is getting faster from Au-Zn to Au-Zn/Ag/Tb. However, the PET time is increasing (meaning slower)! How can a process that requires longer time be a faster process?
9. "Au-Zn NCs was fitted to 30.2 ns composed by 18.4 ns non-radiative (42%)" Why is it non-radiative? It should be a radiative component!
10. The discussions of the tau components and their assignment to various processes seem inconsistent (the part above Fig 4 vs. below Fig 4). For example, the 800-fs component was initially assigned as Sn to S1 (above Fig 4), but it was later assigned to S2 to S1 (below Fig 4 and also in Fig 5).
11. For fs-TA (Fig 5), the plain Au system should be measured and compared with the Zn, Zn/Ag, and Zn/An/Tb systems. The Au system's data may be added to the SI.

Reviewer #3

(Remarks to the Author)

The authors report on the preparation of metal clusters and the influence of the addition of metal salts on the photoluminescence properties. The samples show interesting properties, for example very high quantum yields (how was the quantum yield determined?). However, in the opinion of this reviewer, the work does not meet the necessary high standard to be published.

The main reason is the lack of information on the structure and even composition of the "cluster". Without this, the work remains highly speculative. In addition, the addition of metal ions makes the system a complex one, likely dominated by aggregation. Claims like (in the abstract): "Here, we report efficient suppression of the total structural motions (i.e., vibrations/rotations-dictated structure rigidity) and management of the electron transfer dynamics (i.e., pathways and rates of the radiative relaxation) to boost a near-unity absolute PLQY, by decorating progressive addition of cations." are then difficult to judge (or prove), as many different effects may compete.

Concerning the cluster structure/composition: The authors seem to imply, based on mass spectrometry, that the cluster is very small, four to five metal atoms only. (The masses observed could actually be fragments!) Light scattering on the other hand implies something bigger (a few nm). In any case, the authors are not sure. (What about analysis by gel electrophoresis). The authors are also not sure if the sample is monodisperse, which further complicates the interpretation. In the following the authors analyse their data in the frame of electron – phonon coupling. They talk about optical and acoustic phonons and apply models developed in the field of solid state physics. Talking about (optical and acoustic) phonons for a cluster that has 4 atoms in the core does absolutely not make sense. (Even for a cluster with 1-2 nm core it is not appropriate.)

Version 1:

Reviewer comments:

Reviewer #1

(Remarks to the Author)

The authors have addressed all of my concerns, and the manuscript should be accepted by Nature Communications.

Additionally, I have been asked to assess the response of the authors to the comments of reviewer #3. In my opinion, the authors have well addressed reviewer#3's concerns of structure, PLQY and other issues. Therefore, I recommend it accepted by Nature Communications.

Reviewer #2

(Remarks to the Author)

In the revised manu., the authors have adequately addressed my questions. It is publishable now.

Replies to reviewers' comments and descriptions of revisions made

Reviewer #1 (Remarks to the Author):

General Comments: Luminescent metal nanoclusters with ultra-high photoluminescence quantum efficiency (PLQY) and narrow emission bands are long-pursued for researchers but remain a challenge. This work is very impressive in that the authors report the near-unity PLQY of Au nanoclusters with narrow full width at half maximum (FWHM) of 17-22 nm by the as-proposed sequential cation additives strategy. The progressive introduction of Zn²⁺, Ag⁺, and Tb³⁺ cations strategy is novel and the claimed mechanism of efficient suppression of the total structural motions and management of the electron transfer dynamics is particularly interesting. In addition, the author validated the universality of their strategy, by employing different coin metal architectures and other lanthanide ions. More importantly, the lanthanide ions provide a hopping platform for the excited electrons as their intrinsic ladder-like energy level structure. Overall, this manuscript is well-structured, and the conclusions are well-supported by the presented data. This manuscript meets the high standards of Nature Communications and deserves to be published. Therefore, I strongly recommend accepting this manuscript after solving the following minor issues:

Reply: We sincerely appreciate the reviewer's positive comments on the novelty and significance of our study. We would also like to thank the reviewer for his/her inspiring and constructive comments and suggestions, which have been taken into careful consideration in this revision. Please see below for a point-to-point response to the reviewer's specific comments/suggestions.

Specific Comments:

1. The Zn²⁺ can trigger the enhanced emission of the optical-silenced Au NCs by chelating the COO⁻ of the surface ligand. In addition to Zn²⁺ cations, what are the effects of other cations, such as Ca²⁺ or Mg²⁺, on the photoluminescence boost of the NCs?

Reply: Thank you for this insightful comment. The mechanism of Zn²⁺-triggered self-assembly of Au NCs is assigned to the strong coordination interaction between Zn²⁺ and the deprotonated carboxy group (COO⁻) of MPA ligands on the Au NCs surface (*J. Am. Chem. Soc.* **2021**, 143, 326; *ACS Nano* **2023**, 17, 16644). Following the reviewer's suggestion, we have supplied experiments to assemble Au NCs with other divalent cations (e.g., Ca²⁺ and Mg²⁺). As a result, in contrast to the Zn²⁺-additive system, the solution containing the divalent cations of Ca²⁺ and Mg²⁺ remains clear and transparent even after 24 h incubation (**Figure R1b**). And, there is no obvious photoluminescence (PL) signal detected (**Figure R1b**), suggesting that Ca²⁺ and Mg²⁺ cations have no effects on the self-assembly and PL boosting of Au NCs. The corresponding reasons can be attributed to two aspects: on the one hand, the coordination capacity of Ca²⁺ and

Mg²⁺ is not as strong as that of Zn²⁺ cation. On the other hand, the addition of Zn²⁺ cation can lead to the formation of a hydrogen bond network in Au NCs, therefore generating a new electron transfer pathway and radiative transition (*Sep. Purif. Technol.* **2023**, 310, 122938).

Figure R1. (a) absorption, and (b) PL spectra of individual Au NCs added with divalent cations of Ca²⁺ and Mg²⁺. Insets in (b) are photographs taken under sunlight (left) and 365 nm UV light illumination (right).

2. In Fig. 4, I suggest giving an error bar or error region, and necessary residual and χ^2 in the fitting of the ESA kinetics to judge the fitting quality. In addition, I think the curve's color is too similar in Fig. 4d-f to be identified. I suggest that the authors use some more easily distinguished colors.

Reply: Thank you for your constructive suggestions. As shown in **Figure R2** and **Table R1**, the error region, fitting residual, and reduced χ^2 factor for the excited-state absorption (ESA) kinetics fitting were added. In addition, the curve's color has been changed to be identified clearly.

Figure R2. Comparison of fs-TA spectra and their electron dynamics of serial NCs. (a-c) The fs-TA map pumped at 365 nm. (d-f) The fs-TA profile at different time delays. (g-i) The ESA kinetic decay and fitted residuals within 500 ps (from left to right: Au-Zn, Au-Zn/Ag, and Au-Zn/Ag/Tb NCs).

Table R1. Fitting parameters of normalized TA in **Figure R2**.

Au-based NCs	A_1 ($\times 10^{-3}$)	τ_1 (fs)	A_2 ($\times 10^{-3}$)	τ_2 (ps)	A_3 ($\times 10^{-3}$)	τ_3 (ps)	Reduced χ^2 ($\times 10^{-3}$)
Au-Zn	7.358	800 ± 1.8	12.57	3.1 ± 0.4	10.08	40 ± 0.5	0.19
Au-Zn/Ag	6.759	800 ± 1.5	1.424	3.8 ± 0.3	1.787	27 ± 0.3	1.43
Au-Zn/Ag/Tb	2.441	800 ± 1.6	0.141	5.6 ± 0.4	0.454	12 ± 0.4	1.61

Revisions:

Page 15, Fig. 4:

The fitting curve's color is changed. The fitting residuals were also provided. **Figure R2** has been included as **Fig. 4** in the Manuscript.

Page 15, Line 8-9:

“The ESA kinetic decay and fitted residuals within 500 ps (from left to right: Au-Zn, Au-Zn/Ag, and Au-Zn/Ag/Tb NCs).”

SI, Page 25, Supplementary Table 3:

The error region and reduced χ^2 factor were also provided. **Table R1** has been included as **Supplementary Table 3** in the Supplementary Information.

3. *The authors claimed that “...which is commonly used in semiconductor materials to understand electron-phonon interactions.”, please give the corresponding references.*

Reply: Thank you for this insightful comment. We have added the necessary references in the related sentences (i.e., refs. 51-53).

Revisions:

Page 12, Line 20-23:

“To investigate the additional total structural (surface and interface) vibrations of the NCs, we focus on temperature-dependent emission broadening in > 150 K, which is commonly used in semiconductor materials to understand electron-phonon/electron-vibration interactions⁵¹⁻⁵³.”

Page 25, Line 6-12:

“51. Rudin, S., Reinecke, T. L. & Segall, B. Temperature-dependent exciton linewidths in semiconductors. *Physical Review B* **42**, 11218-11231 (1990).

52. Alivisatos, A. P., Harris, A. L., Levinos, N. J., Steigerwald, M. L. & Brus, L. E. Electronic states of semiconductor clusters: Homogeneous and inhomogeneous broadening of the optical spectrum. *J. Chem. Phys.* **89**, 4001-4011 (1988).

53. Wright, A. D. et al. Electron-phonon coupling in hybrid lead halide perovskites. *Nat. Commun.* **7**, No. 11755 (2016).”

4. *Before publication, details regarding the experimental conditions for MALDI-MS, such as the matrix, the ratio of Au NCs to matrix, and the instrument, should be included in the Supporting Information.*

Reply: Thank you for this insightful comment. Details regarding the experimental conditions for matrix-assisted laser desorption ionization-time-of-flight (MALDI-TOF) have been added to the Supplementary Information file.

Revisions:

SI, Page 5, Line 11-18:

“Matrix-assisted laser desorption ionization-time-of-flight (MALDI-TOF) mass spectra were tested on Bruker Autoflex speed TOF under a positive linear mode.

Trans-2-[3-(4-tert-Butylphenyl)-2-methyl-2-propenyldiene] malononitrile (DCTB) was employed as the matrix for all samples. Specific details are described as follows: The 10 μL DCTB matrix and 10 μL NCs aqueous solution were fully mixed. Then, spot 1.5 μL of the mixed solution on the target plate and wait for the solvent to evaporate. The detector gain was set at 10. The spot size was fixed at medium, and the laser intensity was tuned to be greater than 70 %.”

5. *In addition to the chelation of Tb^{3+} with the surface ligands of NCs, the Tb^{3+} also provides a hopping platform for the excited electrons from the staple motif to the metal core. How can the authors prove that the energy level of Tb^{3+} is located between the S_n of the staple motif and the S_2/S_1 energy levels of the metal core?*

Reply: Thank you for this insightful comment. 1) The S_1 position can be generally estimated by transforming the emission peak (for those emissive states) or the absorption peak (for those non-emissive states) (*Nat. Chem.* **2020**, 12, 345; *Chem. Sci.* **2019**, 10, 10170). In our system, according to the emission peak ($S_1 \rightarrow S_0$), the S_1 energy level of the metal core in Au-Zn, Au-Zn/Ag, Au-Zn/Ag/Tb NCs are calculated to be 20.0×10^3 , 20.4×10^3 , and $20.4 \times 10^3 \text{ cm}^{-1}$, respectively. In parallel, the S_1 energy level position of the staple motif can be estimated to be 28.2×10^3 , 29.7×10^3 , and $29.1 \times 10^3 \text{ cm}^{-1}$, by considering the characteristic absorption peaks ($S_0 \rightarrow S_1$) of the staple motif in Au-Zn (355 nm), Au-Zn/Ag (337 nm), and Au-Zn/Ag/Tb (344 nm), respectively. 2) The inherent $^5\text{D}_3$ energy level of Tb^{3+} is located at $26.3 \times 10^3 \text{ cm}^{-1}$ (*Chem. Rev.* **2022**, 122, 5519; *Light: Sci. Appl.* **2016**, 5, e16066). Thus, the energy level of Tb^{3+} is between the S_1 level of the staple motif and the S_1 level of the metal core. In this way, Tb^{3+} ions serve as an energy bridge to promote the PET process. Accordingly, we have modified the energy level diagram to clarify the S_1 location to give a clearer presentation.

Reviewer #2 (Remarks to the Author):

General Comments: In this manuscript, Wang et al. synthesized 3-mercaptopropionic acid (MPA) protected Au nanoclusters and introduced Zn^{2+} , Ag^+ and rare earth Tb^{3+} ions to improve the PL intensity of the as-synthesized nanoclusters. MALDI-MS, XPS and DLS were employed to understand the composition of the nanoclusters. Low-temperature PL analysis and TAS were also used to study the electron dynamic and PL mechanism. This work is interesting and suitable for publication in Nat. Commun. after addressing the following questions.

Reply: We are glad that the reviewer finds this work interesting and important. We also appreciate the professional comments/suggestions provided by the reviewer, which have spurred improvements in both the readability and scientific content of our manuscript. These comments/suggestions have been taken into careful consideration and a point-to-point response can be found in the coming paragraphs. We hope the quality of our revised manuscript has been greatly improved according to the reviewer's

comments/suggestions and can be considered for publication in *Nature Communications*.

Specific Comments:

1. Line 173, it would be better to use meV to describe FWHM. Because, on the energy scale, 22 nm linewidth in the blue region is much larger than the same 22 nm linewidth in the NIR region.

Reply: Thanks for the reviewer's professional comment. As the reviewer suggested, we have modified the unit of FWHM and Stokes Shift in the revised Supplementary Information file.

Table R2. Optical parameters of Au-Zn, Au-Zn/Ag, and Au-Zn/Ag/Tb NCs.

Gold NCs	Absolute PLQY (%)	FWHM (meV)	Stokes shift (meV)
Au-Zn	51.2	86	61
Au-Zn/Ag	83.4	108	85
Au-Zn/Ag/Tb	99.5	99	75

Revisions:

Page 6, Line 18-20:

“In addition, these serial NCs manifest narrow full width at half maximum (FWHM) of 86-108 meV, and a small Stokes shift of 61-85 meV (**Supplementary Table 1**).”

SI, Page 25, Supplementary Table 1:

Table R2 has been included as **Supplementary Table 1** in the Supplementary Information.

2. In this work, the analysis of the composition of the nanoclusters heavily relies on the MALDI-MS results. In fig. s6a, there is a strong peak on the left side of the 1147.3 peak and a weak peak on the right side of the 1147.3 peak, please assign them.

Reply: Thank you for this insightful comment. A strong peak on the left side of the 1147.3 peak is located at 1083 Da, which is assigned to the signal of [Au₄(MPA)₂+Na+Zn-2H] fragments. A weak peak (1211.7 Da) on the right side of the 1147.3 peak is due to the adsorption of Zn²⁺ on the intact [Au₄(MPA)₃+2Na-2H] NCs and can be attributed to the [Au₄(MPA)₃+2Na+Zn-2H].

Revisions:

SI, Page 12, Line 7-11:

“In Supplementary Fig. 6a, A strong peak at 1083 Da is attributed to the signal of [Au₄(MPA)₂+Na+Zn-2H] fragment. A weak peak (1211.7 Da) on the right side of the 1147.3 peak is due to the adsorption of Zn²⁺ on the [Au₄(MPA)₃+2Na-2H] NCs and can be attributed to the [Au₄(MPA)₃+2Na+Zn-2H].”

3. Line 213, the author mentioned the addition of Tb³⁺ affects the electron density of silver. Why are Au atoms not affected? If the mechanism is correct, Au should also be affected because it's the majority element in the cluster.

Reply: Thank you for this insightful comment. We are sorry that there was an improper correction of the C 1s X-ray photoelectron spectra (XPS) signal in our first submission. To solve it, we have checked the original XPS data and recalibrated the binding energy of C, Au, and Ag elements. The corresponding corrected XPS results are shown in **Figure R3**. The Au 4f_{7/2} peak in Au-Zn NCs is located at 84.42 eV. After stepwise introducing Ag⁺ and Tb³⁺, the Au 4f_{7/2} peaks in resultant Au-Zn/Ag (84.54 eV) and Au-Zn/Ag/Tb NCs (84.65 eV) exhibit positive shifts of 0.12 and 0.11 eV, respectively. Meanwhile, as shown in **Figure R3b**, the Ag 3d_{5/2} peaks in Au-Zn/Ag and Au-Zn/Ag/Tb NCs are detected at 367.89 eV and 368.03 eV, respectively. There are 0.14 eV positive shifts of Ag 3d_{5/2} peaks after Tb³⁺ addition. Positive shifts of the binding energy can be rationally assigned to the reduced electron density around Au and Ag, which may be determined reasonably, by the change of the electron distribution of the NCs due to the addition of Ag⁺, or the coordination interaction between Tb³⁺ and COO⁻ on the surface of the NCs (*Small* **2023**, 19, 2301357; *Adv. Optical Mater.* **2023**, 11, 2300151).

In addition, we also retested the high-resolution Au 4f and Ag 3d XPS spectra of Au-Zn, Au-Zn/Ag, and Au-Zn/Ag/Tb NCs (**Figure R4**). As expected, the binding energy of Au 4f and Ag 3d show positive shifts with the introduction of metal cations. In concrete terms, the Au 4f_{7/2} peaks exhibit positive shifts of 0.11 and 0.12 eV after progressively introducing Ag⁺ and Tb³⁺, respectively. Similarly, the Ag 3d_{5/2} peaks exhibit positive shifts of 0.14 eV after introducing Tb³⁺. These additional results prove that the addition of metal cations can indeed affect the total electron density of NCs.

Figure R3. (a, b) High-resolution Au 4f XPS and Ag 3d XPS spectra of a series of NCs.

Figure R4. (a, b) High-resolution Au 4f XPS and Ag 3d XPS spectra of Au-Zn, Au-Zn/Ag, and Au-Zn/Ag/Tb NCs, respectively.

Revisions:

Page 8, Line 3-16:

“The Au 4f_{7/2} peak in Au-Zn NCs (84.42 eV) is between Au(0) NPs (84.15 eV) and Au(I)-p-MBA complex (85.00 eV, Fig. 2a), indicating that Au-Zn NCs formed a classical core-shell structure including Au(0) core and Au(I)-S staple motif. After progressively introducing Ag⁺ and Tb³⁺, the Au 4f_{7/2} peaks in Au-Zn/Ag (84.54 eV) and Au-Zn/Ag/Tb NCs (84.65 eV) exhibit positive shifts of 0.12 and 0.11 eV, respectively. Meanwhile, the Ag 3d_{5/2} peaks in Au-Zn/Ag (367.89 eV) and Au-Zn/Ag/Tb (368.08 eV) NCs are between Ag(0) NPs (367.35 eV) and Ag(I)-p-MBA complexes (368.65 eV), suggesting that the existence of Ag(0) and Ag(I) species in the metal core and staple motifs, respectively (Fig. 2b). There are 0.14 eV positive shifts of Ag 3d_{5/2} peaks after Tb³⁺ addition. Positive shifts of the binding energy can be rationally assigned to the reduced electron density around Au and Ag, which is determined by many factors, such as the change of the electron distribution of the NCs due to the addition of Ag⁺, or the coordination interaction between Tb³⁺ and COO⁻ on the surface of the NCs^{39, 41}.”

Page 10, Fig. 2:

The binding energy of C, Au, and Ag have been recalibrated. **Figure R3a, b** has been included as **Fig. 2a, b** in the Manuscript.

4. Line 223, the author mentioned “the added Ag⁺ serves as a linker to bridge the small Au(I)-thiolate motifs on the parental NCs surface to increase the length and rigidity of the staple motifs”. So, how does Ag⁺ serve as the linker? And, why is Ag⁺ itself not enough (“isolated AuAg NCs with non-photoluminescence”) to make the MPA-protected nanocluster emissive that Zn must be introduced?

Reply: We are sorry that we didn’t articulate well about the role of Ag⁺ cations. Thiolate-protected gold NCs can be regarded as ultrasmall (< 3 nm) nanoparticles. They

feature the unique metal(0)@metal(I)-ligand core-shell total structure to construct a “divide-and-protect” model, that is, the “staple motif” of metal(I)-ligand wrapping dynamically over the metal(0) core. Conventional metal NCs have short metal(I)-thiolate motifs (*J. Am. Chem. Soc.* **2012**, 134, 16662). Through precise doping strategies, heteroatoms can be doped into the metal core, interfacial staple motif, or both two positions (*Angew. Chem. Int. Ed.* **2014**, 53, 2376; *J. Am. Chem. Soc.* **2019**, 141, 5314; *J. Am. Chem. Soc.* **2018**, 140, 14235). There are two cases in which heteroatoms are doped on the pinning primitives at the cluster interface: one is that heteroatoms replace Au atoms in the staple motif (*J. Am. Chem. Soc.* **2017**, 139, 17779), and the other is that Ag^+ is inserted into the motif without replacing any atoms (*Nanoscale* **2014**, 6, 157). To further clarify the role of Ag^+ in the staple motif, we have performed additional XPS analyses of serial NCs. The ratio of Au(I)/Au(0) was calculated by deconvoluting the Au $4f_{5/2}$ and $4f_{7/2}$ XPS signals, which is 2.963:1, 2.965:1, and 2.957: 1 for the Au-Zn, Au-Zn/Ag, and Au-Zn/Ag/Tb NCs (**Figure R5**), respectively. These results demonstrate that the proportion of Au(I) to Au(0) remains constant in different NCs. In addition, according to the molecular formula of the NCs (i.e., $[\text{Au}_4(\text{MPA})_3+2\text{Na}-2\text{H}]$, $[\text{Au}_4\text{Ag}(\text{MPA})_3+3\text{Na}-3\text{H}]$, and $[\text{Au}_4\text{Ag}(\text{MPA})_3+3\text{Na}-3\text{H}]$ for the Au-Zn, Au-Zn/Ag, and Au-Zn/Ag/Tb NCs, respectively), the content of Au remains unchanged with the addition of metal cations. Taking these XPS and MALDI-TOF mass spectra results into combined consideration, Ag^+ is more likely to be inserted into the motif instead of replacing pristine Au atoms in that. This doping mode can lead to the great arrangement of staple motifs, that is, Ag^+ acts as linkers to connect short Au(I)-thiolate motifs into long Au(I)/Ag(I)-thiolate and increase the rigidity of the staple motifs (*Nanoscale* **2014**, 6, 157).

It should be mentioned that the added Ag^+ only serves as linkers on the intra-nanocluster to bridge the small Au(I)-thiolate motifs, while the added Zn^{2+} cations mainly work to trigger the inter-nanocluster assembly. This Zn^{2+} -induced assembly process is driven by electrostatic interactions between deprotonated carboxyl of MPA ligands in NCs and Zn^{2+} cations, resulting in the enhancement of initial emission intensity in leveraging the aggregation-induced emission effect. Therefore, the collective effects of intra-NCs (Ag^+) and inter-NCs (Zn^{2+}) aggregation endow NCs with ultrahigh PL performance.

Figure R5. (a-c) High-resolution Au $4f$ XPS fitting of Au-Zn, Au-Zn/Ag, and Au-Zn/Ag/Tb NCs, respectively.

5. Line 235, the author claimed “Adding Zn^{2+} to Au NCs significantly increases the average size from 2.47 to 270 nm”. Is it possible to observe such a size increase under TEM?

Reply: Thank you for this insightful comment. As suggested by the reviewer, we have carried out additional transmission electron microscopy (TEM) measurements to give more evidence of the size variation of NCs. As shown in **Figure R6a**, the individual Au NCs feature an average diameter of 2.47 ± 0.44 nm with a good monodispersity. After adding Zn^{2+} , Ag^+ , and Tb^{3+} into the isolated Au NCs, a series of NCs show irregular aggregation. The size of a series of clusters is well consistent with the results of dynamic light scattering (DLS) (i.e., 230, 270, and 360 nm for Au-Zn, Au-Zn/Ag, and Au-Zn/Ag/Tb NCs, respectively). In addition, the high-resolution TEM images show that Au-Zn, Au-Zn/Ag, and Au-Zn/Ag/Tb NCs are assembled by the individual NCs with an average diameter of 2.50 ± 0.42 nm, 2.52 ± 0.42 , and 2.52 ± 0.50 nm, respectively (**Figure R6b-d**).

Figure R6. (a-d) TEM images of Au, Au-Zn, Au-Zn/Ag, and Au-Zn/Ag/Tb NCs. The insets are amplified images of the corresponding samples.

Revisions:

Page 9, Line 7-8:

“...we performed dynamic light scattering (DLS) and transmission electron microscopy (TEM) measurements.”

Page 9, Line 12-13:

“TEM gives more evidence of the size growth of NCs, as shown in **Supplementary**

Fig. 8.”

SI, Page 5, Line 21-23:

“TEM images were performed on the FEI Tecnai G2 F20 microscopy operated at 200 kV.”

SI, Page14, Supplementary Fig. 8:

Figure R6 has been included as **Supplementary Fig. 8** in the Supplementary Information.

6. *Were the "300 to 20 K" measurements done on solutions or films? No experimental detail can be found.*

Reply: Thank you for this insightful comment. The temperature-dependent (300 to 20 K) PL of serial NCs was measured on the film. To avoid significant changes in the NCs structure and the corresponding PL properties due to the freezing of the aqueous solution at low temperatures, the film samples were used for the temperature-dependent PL measurements.

Revisions:

SI, Page 6, Line 14-16:

“Temperature-dependent PL spectra of NCs film were measured on FLS1000 with an Oxford Optistat attemperator cooled by liquid nitrogen.”

7. *Line 406, the author mentioned “the excited state electrons in staple motifs spontaneously transfer into the S₂ excited state of the metal core due to the minimum energy discrepancy”. Is there any evidence that supports this energy argument or is it just a guess? Similarly, line 421, “The ⁵D₃ level of Tb³⁺ is located between the S_n level of the staple motif and the S₂ level of the metal core”.*

Reply: Thank you for this insightful comment. 1) We are sorry for the unclear expression in this sentence (i.e., “The excited state electrons in staple motifs spontaneously transfer into the S₂ excited state of the metal core due to the minimum energy discrepancy”). Herein, what we would like to express is that the excited state electrons in staple motifs spontaneously transfer into the S₁ or S_n energy level of the metal core which is closest to the S₁ or S_n energy level of staple motifs. To eliminate any confusion, we have deleted such a sentence in the revised manuscript. 2) We are sorry for the confusing statement in “The ⁵D₃ level of Tb³⁺ is located between the S_n level of the staple motif and the S₂ level of the metal core”, we have strictly corrected the definition of energy level. That is, “The ⁵D₃ level of Tb³⁺ is located between the S₁ level of the staple motif and the S₁ level of the metal core”. Because, on the one hand, the S₁ position can be in general estimated by transforming the emission peak (for those emissive states) or the absorption peak (for those non-emissive states) (*Nat. Chem.* **2020**,

12, 345; *Chem. Sci.* **2019**, 10, 10170). In our system, according to the emission peak ($S_1 \rightarrow S_0$), the S_1 energy level of the metal core in Au-Zn, Au-Zn/Ag, Au-Zn/Ag/Tb NCs are calculated to be 20.0×10^3 , 20.4×10^3 , and 20.4×10^3 cm^{-1} , respectively. In parallel, the S_1 energy level position of the motif can be estimated to be 28.2×10^3 , 29.7×10^3 , and 29.1×10^3 cm^{-1} , by considering the characteristic absorption peaks ($S_0 \rightarrow S_1$) of the staple motif in Au-Zn (355 nm), Au-Zn/Ag (337 nm), and Au-Zn/Ag/Tb (344 nm), respectively. On the other hand, the inherent 5D_3 energy level of Tb^{3+} is located at 26.3×10^3 cm^{-1} (*Chem. Rev.* **2022**, 122, 5519; *Light: Sci. Appl.* **2016**, 5, e16066). Thus, the energy level of Tb^{3+} is between the S_1 level of the staple motif and the S_1 level of the metal core. Accordingly, we have modified the energy level diagram to clarify the S_1 location to give a clearer presentation.

8. *The author claimed electron transfer from the nanocluster to Tb^{3+} and back to the nanocluster, which is not convincing. In Fig. 5, the author claimed PET is getting faster from Au-Zn to Au-Zn/Ag/Tb. However, the PET time is increasing (meaning slower)! How can a process that requires longer time be a faster process?*

Reply: We sincerely appreciate the reviewer for your professional and constructive comments. 1) In our reply to the “Comment 7” and the corresponding corrections, we have clarified that the calculation of the S_1 energy level of the core state and the staple motif for a series of NCs (i.e., S_1 energy level of the core state: 20.0×10^3 , 20.4×10^3 , and 20.4×10^3 cm^{-1} for Au-Zn, Au-Zn/Ag, Au-Zn/Ag/Tb NCs; S_1 energy level of the staple motif: 28.2×10^3 , 29.7×10^3 , and 29.1×10^3 cm^{-1} for Au-Zn, Au-Zn/Ag, Au-Zn/Ag/Tb NCs). The intrinsic 5D_3 energy level of Tb^{3+} is 26.3×10^3 cm^{-1} (*Chem. Rev.* **2022**, 122, 5519; *Light: Sci. Appl.* **2016**, 5, e16066). In such a scenario, the electron transfer pathway, from the NCs to Tb^{3+} and back to the nanocluster, is much more convincing. It means that the 5D_3 level of Tb^{3+} is between the S_1 energy level of the core state and that of the staple motif in the NCs, which serves as a hopping platform for the electron transfer of excited electrons as their intrinsic ladder-like energy level structure. 2) We share your viewpoint that longer PET times mean slower electron transfer rates. In our first submission, it is improper for the ESA kinetic fitting because the data acquisition time is too long and the source of the excited-state absorption (ESA) signal is not clear. To solve this problem, pumping energy-dependent femtosecond-transient absorption (fs-TA) measurements have been implemented. As shown in **Figure R7**, the photo-excitation with a 2.76 eV (450 nm) laser pulse only excited the ground-state electrons of the metal core in the metal NCs, no ground-state bleaching (GSB), and ESA signal is identified in Au-Zn, Au-Zn/Ag, and Au-Zn/Ag/Tb NCs. This proves that the ESA signal generated by the 3.40 eV (365 nm) laser pulse originates from the staple motif (the electron donor) instead of the metal core (the electron acceptor) in the NCs. When the pump energy is gradually increased from 2.92 eV (425 nm) to 3.40 eV (365 nm), the ESA signal from the staple motif is observed and gradually enhanced (**Figure R8-10**). In addition, we have also retested and fitted fs-TA of Au-Zn, Au-Zn/Ag, and Au-Zn/Ag/Tb NCs. In concrete terms, as shown in **Figure R10**, upon photo-excitation with a 3.40 eV (365 nm) laser pulse, an apparent GSB signal can be identified at 443 nm in

the TA spectra of Au-Zn NCs. After incorporating Ag⁺ and Tb³⁺, their GSB peaks position blue shifts to 432 and 429 nm for Au-Zn/Ag and Au-Zn/Ag/Tb NCs, respectively. Following their GSB band, a broad ESA band from the staple motif of the NCs is observed, manifesting the dense excited states of serial NCs (*J. Am. Chem. Soc.* **2020**, 142, 18086). Within an approximate 500 ps time window, three decay components were fitted out in the ESA kinetics: 800 fs, 3.1 ps, and 40 ps for Au-Zn NCs; 800 fs, 3.8 ps, and 27 ps for Au-Zn/Ag NCs; 800 fs, 5.6 ps, and 12 ps for Au-Zn/Ag/Tb NCs, respectively. The time component of dozens of picoseconds processes (40, 27, and 12 ps) are relevant to the PET process from the excited state of staple motifs to the metal core. The gradual decrease of this time component confirms that the PET process gets faster with the sequential addition of metal cations (i.e., Ag⁺ and Tb³⁺). In addition, we have also retested the fs-TA maps of Au-Zn/Ag/R NCs added with different rare earths and further fitted their corresponding ESA decay kinetics at 475 nm in a 500 ps time scale (**Figure R11** and **Table R3**). It was found that the PET time constant of rare-earth ions-contained NCs relates to the discrepancy in the energy level position of different rare-earth ions. The ideal rare earth ions to act as a PET bridge are the ones whose energy level should lie between the S₁ level of the staple motif and the S₁ level of the metal core. Finally, we have redrawn the PET time diagram of the Au-Zn/Ag and Au-Zn/Ag/R (**Figure R12**).

Figure R7. Comparison of fs-TA spectra and their electron dynamics of serial NCs. (a-c) The fs-TA map pumped at 450 nm (The blank of 510-575 nm has green light components from the pump laser). (d-f) The ESA kinetic curve within 500 ps (from left to right: Au-Zn, Au-Zn/Ag, and Au-Zn/Ag/Tb NCs).

Figure R8. Comparison of fs-TA spectra and their electron dynamics of serial NCs. (a-c) The fs-TA map pumped at 425 nm (The blank of 535-565 nm has green light components from the pump laser). (d-f) The ESA kinetic curve within 500 ps (from left to right: Au-Zn, Au-Zn/Ag, and Au-Zn/Ag/Tb NCs).

Figure R9. Comparison of fs-TA spectra and their electron dynamics of serial NCs. (a-c) The fs-TA map pumped at 400 nm. (d-f) The ESA kinetic curve within 500 ps (from left to right: Au-Zn, Au-Zn/Ag, and Au-Zn/Ag/Tb NCs).

Figure R10. Comparison of fs-TA spectra and their electron dynamics of serial NCs. (a-c) The fs-TA map pumped at 365 nm. (d-f) The fs-TA profile at different time delays. (g-i) The ESA kinetic decay and fitted residuals within 500 ps (from left to right: Au-Zn, Au-Zn/Ag, and Au-Zn/Ag/Tb NCs).

Figure R11. (a, c) The fs-TA maps of Au-Zn/Ag/R NCs upon 365 nm laser excitation. (b, d) The kinetic decay at 475 nm of Au-Zn/Ag/R NCs within 500 ps. “R” is for Ce^{3+} , Pr^{3+} , Nd^{3+} , Sm^{3+} , Eu^{3+} , Dy^{3+} , Ho^{3+} , Er^{3+} , Tm^{3+} , and Yb^{3+} , respectively.

Figure R12. The comparison of PET between Au-Zn/Ag and Au-Zn/Ag/R NCs.

Table R3. Fitting parameters of normalized TA of Au-Zn/Ag/R NCs.

Rare earth	$A_1(\times 10^{-3})$	τ_1 (fs)	$A_2(\times 10^{-3})$	τ_2 (ps)	$A_3(\times 10^{-3})$	τ_3 (ps)
Tb ³⁺	2.441	800	0.141	5.6	0.454	12
Sm ³⁺	2.417	800	0.149	5.6	0.741	13
Dy ³⁺	2.881	800	0.115	5.6	0.984	14
Nd ³⁺	2.670	799	0.230	5.2	0.983	16
Er ³⁺	2.216	800	0.128	5.0	0.595	17
Eu ³⁺	2.637	801	0.116	4.8	0.274	18
Pr ³⁺	2.915	800	0.115	4.7	0.938	19
Ho ³⁺	2.432	801	0.197	4.3	0.349	21
Tm ³⁺	2.911	800	0.460	4.2	0.167	22
Ce ³⁺	2.666	800	0.720	4.0	0.207	24
Yb ³⁺	3.253	799	0.183	3.8	0.222	26
Gd ³⁺	2.515	800	0.322	3.8	0.317	27

Revisions:

Page 2, Line 12-14:

“Moreover, introducing cation additives significantly reduces electron transfer time (from 40, 27 to 12 ps) in the pathway from staple motifs to the metal core.”

Page 14, Line 11-13:

“Following their GSB band, a broad excited-state absorption (ESA) band contributing by staple motif is observed, manifesting the dense excited states of serial NCs⁵⁴.”

Page 25, Line 13-15:

“54. Wang, J.-Q. et al. Total structure determination of the largest alkynyl-protected fcc gold nanocluster Au₁₁₀ and the study on its ultrafast excited-state dynamics. *J. Am. Chem. Soc.* **142**, 18086-18092 (2020).”

Page 14, Line 18-21:

“Within an approximate 500 ps time window, three decay components were fitted out in the ESA kinetics: 800 fs, 3.1 ps, and 40 for Au-Zn NCs; 800 fs, 3.8 ps, and 27 ps for Au-Zn/Ag NCs; 800 fs, 5.6 ps, and 12 ps for Au-Zn/Ag/Tb NCs (**Fig. 4g-i and Supplementary Table 3**).”

Page 14, Line 25-28:

“The several picosecond time components (3.1, 3.8, and 5.6 ps) are assigned to the core-directed structural vibration in serial NCs⁴⁹. Finally, the relatively slow decay process (40, 27, and 12 ps) can be relevant to the PET process from the excited state of staple motifs to the metal core^{15,43,58}.”

Page 14, Line 30 to Page 15, Line 3:

“Intriguingly, the time component of dozens of picoseconds gradually decreases. This can be explained by the fact that the excited state electrons from the staple motif transfer to the metal core faster.”

Page 15, Line 8-9:

“The ESA kinetic decay and fitted residuals within 500 ps (from left to right: Au-Zn, Au-Zn/Ag, and Au-Zn/Ag/Tb NCs).”

Page 15, Fig. 4:

Figure R10 has been included as **Fig. 4** in the Manuscript.

Page 18, Fig. 6:

The fitting PET times of Au-Zn/Ag/R NCs have been revised in **Figure R12**. **Figure R12** has been included as **Fig. 6b** in the Manuscript.

Page 18, Table 1:

The fitting parameters of normalized TA of Au-Zn/Ag/R NCs have been revised. **Table R3** has been included as **Table 1**.

SI, Page 20, Supplementary Fig. 14:

Figure R11 has been included as **Supplementary Fig. 14** in Supplementary Information.

9. "Au-Zn NCs was fitted to 30.2 ns composed by 18.4 ns non-radiative (42%)" Why is it non-radiative? It should be a radiative component!

Reply: Thank you for this insightful comment. It is generally accepted that the nonradiative relaxation in metal NCs is caused by the electron-vibration coupling associated with lattice thermal effects. The proportion of nonradiative relaxation should gradually decline with the decrease of environmental temperature due to the suppressed vibration behavior at low temperatures (*J. Am. Chem. Soc.* **2023**, 145, 19969; *ACS Nano*

2024, 18, 21534). To prove the short lifetime ($\tau_1 = 18.4$ ns) is a nonradiative process in Au-Zn NCs, we carried out temperature-dependent time-resolved photoluminescence (TRPL) measurements. As shown in **Figure R13**, with the temperature decreasing from 300 to 20 K, it is obvious that the lifetime becomes longer and the decay traces gradually convert from biexponential to monoexponential. We further fitted all the decay traces and found that the proportion of τ_1 gradually declined with the decrease in temperature and even disappeared at 20 K (**Table R4**). These results present that τ_1 can be reasonably assigned to non-radiative relaxation, while τ_2 belongs to radiative relaxation.

Figure R13. The temperature-dependent TRPL decay traces of Au-Zn NCs upon 375 nm excitation. The applied temperature range is 300-20 K and the temperature interval between two adjacent records is 40 K.

Table R4. Fluorescent lifetime components of Au-Zn NCs obtained from TCSPC measurements.

Temperature (K)	τ_1 (ns)	Proportion (%)	τ_2 (ns)	Proportion (%)	$\tau_{ave.}$ (ns)	χ^2
300	21.4	41	49.9	59	38.1	1.4000
260	21.9	37	63.0	63	47.7	1.2401
220	22.3	29	61.5	71	50.1	1.2560
180	23.2	23	62.3	77	53.4	1.2797
140	23.7	18	69.8	82	61.7	1.5202
100	24.4	11	87.1	89	80.3	1.2974
60	24.7	5	102.1	95	98.6	1.2303
20	-	-	116.0	100	116.0	1.2150

10. The discussions of the tau components and their assignment to various processes seem inconsistent (the part above Fig 4 vs. below Fig 4). For example, the 800-fs component was initially assigned as S_n to S_1 (above Fig 4), but it was later assigned to S_2 to S_1 (below Fig 4 and also in Fig5).

Reply: Thank you for this insightful comment. We are sorry for the confusion due to the inconsistent phrasing in our first submission. To avoid this, we have modified the

related sentences and diagram (**Figure R14** and **Figure R15**) in the revised manuscript throughout (that is “ S_n to S_1 ” but not “ S_2 to S_1 ”) to clarify the photophysical mechanism.

Figure R14. Schematic diagram illustrating the excited-state dynamics of serial NCs. Arrows denote the transitions between different electronic states. S_0 : ground state, S_1 : lowest singlet state, S_n : high singlet state, IC: internal conversion; SR: structural relaxation; PET: photoinduced electron transfer.

Figure R15. Schematic illustration of rare earth ions acting as an electron transfer bridge and energy level diagram of rare earth ions. The reddish-brown shaded area indicates that the rare earth can act as an electron transfer bridge, while the rare earth ions in the gray-shaded area with a cross sign cannot.

Revisions:

Page 15, Line 15 to Page 16, Line 5:

“Then, the excited state electrons in staple motifs spontaneously transfer into the S_n excited state of the metal core. Subsequently, they experience a fast decay into the S_1 state of the metal core through the IC process ($S_n \rightarrow S_1$). After that, the excited electrons in the S_1 state non-radiatively relax into the lowest energy level of S_1 through structural relaxation. Finally, the radiative relaxation of excited-state electrons from the S_1 to the S_0 ground state in the metal core can emit ~ 490 nm fluorescence with a ns -level lifetime.”

Page 16, Line 14-16:

“The 5D_3 level of Tb^{3+} is located between the S_1 level of the staple motif and the S_1 level of the metal core, which serves as a bridge to promote the PET process.”

Page 17, Line 4-7:

“Changes in the time constant of PET, SR, and coupling strength of electrons-high-frequency vibration from NCs surface and interface. S_0 : ground state, S_n : high singlet state, S_1 : lowest singlet state, IC: internal conversion; SR: structural relaxation; PET: photoinduced electron transfer.”

Page 17, Fig. 5; Page 18, Fig. 6:

We have modified the phrasing of “ S_2 ” to “ S_n ” in **Figure R14** and **Figure R15**. **Figure R14** and **Figure R15** have been included as **Fig. 5a** and **Fig. 6d** in the Manuscript.

11. For fs-TA (Fig 5), the plain Au system should be measured and compared with the Zn, Zn/Ag, and Zn/Ag/Tb systems. The Au system's data may be added to the SI.

Reply: Thank you for this insightful comment. The absorption region of Au NCs is mainly below 300 nm, while the pump wavelength of the femtosecond transient absorption spectrometer is limited to greater than 320 nm. The pump light of > 300 nm cannot efficiently excite Au NCs. Based on this fact, to compare the fs-TA signals of Au NCs with that of Zn, Zn/Ag, and Zn/Ag/Tb systems, the 365 nm laser pulse is adopted to excite Au NCs. As shown in **Figure R16**, no GSB or ESA signal was detected in the fs-TA map of Au NCs.

Figure R16. The fs-TA spectra and electron dynamics of isolated Au NCs. (a) The fs-TA map pumped at 365 nm. (b) The fs-TA profile at different time delays. (c) The kinetic decay curve.

Revisions:

SI, Page 18, Supplementary Fig. 12:

Figure R16 has been included as **Supplementary Fig. 12** in Supplementary Information.

Reviewer #3 (Remarks to the Author):

General Comments: The authors report on the preparation of metal clusters and the influence of the addition of metal salts on the photoluminescence properties. The

samples show interesting properties, for example very high quantum yields... However, in the opinion of this reviewer, the work does not meet the necessary high standard to be published.

Reply: We are grateful to this reviewer for his/her professional insights in reviewing our manuscript. We would also like to thank the reviewer for his/her inspiring and constructive comments and suggestions, which have spurred improvements in both the readability and scientific content of our manuscript. We hope our following point-to-point response and corresponding revisions are satisfactory so that the revised manuscript can meet the high standards and be considered for publication in *Nature Communications*.

Specific Comments:

1. *The samples show interesting properties, for example very high quantum yields (how was the quantum yield determined?).*

Reply: Thank you for this insightful comment. We're glad that the reviewer agrees with the importance of our research topic. Indeed, optimizing the quantum yields of luminescent metal NCs is vital not only to understanding the structure-property relationship but also to increasing the acceptance of metal NCs as novel luminescent materials for multiple applications. However, there is still a very limited success in attaining high quantum yields (commonly below 30%) as the complexity, diversity, and mutability in terms of radiative relaxation of electron dynamics (e.g., the pathways, delay time, and trap states) (*J. Am. Chem. Soc.* **2015**, 137, 12906; *Nature Chem.* **2017**, 9, 689; *Angew. Chem. Int. Ed.* **2019**, 58, 8139; *Chem. Soc. Rev.* **2021**, 50, 2297; *Nat. Commun.* **2023**, 14, 658; *J. Am. Chem. Soc.* **2024**, 146, 22180). In this work, we achieved efficient suppression of total structural vibrations and management of electron transfer dynamics toward achieving remarkable enhancement in emission intensity (up to near-unity PLQY of 99.5%), by elaborately decorating the sequential addition of different metal cations (e.g., Zn²⁺, Ag⁺, and Tb³⁺) in the Au NCs. The underlying photophysical mechanism was systematically investigated by using serial steady- and transient-state optical characterizations. On the one hand, the sequential addition of metal cations can decrease the vibration frequency of the metal core, surface motifs, and terminal ligand groups. These effects further collectively suppress the non-radiative structural vibration of the heterogeneous metal NCs. On the other hand, the sequential metal cations addition is more conducive to the electron transfer from the staple motifs to the metal core, which significantly boosts the metal core-related emission up to a near-unity PLQY at room temperature in an aqueous solution. Our findings give new insights for the optimization of PL properties of metal NCs, which would increase their acceptance as novel luminescent materials and promote their practical applications.

The PLQY of all samples in aqueous solution was measured and calculated on a QM8000 HORIBA spectrometer attached with an integrating sphere coating with a reflective BaSO₄ layer. For the detailed measurements of the PLQY of all samples, an Xe lamp with a fixed emission wavelength at 365 nm was used as the excitation source.

Pure water support was measured first in the integrating sphere and used as blank references for the samples in an aqueous solution. The calculation of absolute PLQY values was conducted on the built-in “FelixGX” software (version 4.9.0.10321). The values of PLQY were calculated by using following equation: $PLQY = \frac{\int \lambda P(\lambda) d\lambda}{\int \lambda \{E(\lambda) - R(\lambda)\} d\lambda}$, where $E(\lambda)/h\nu$, $R(\lambda)/h\nu$, and $P(\lambda)/h\nu$ are the number of photons in the spectrum of excitation, reflectance, and emission, respectively. Using the method mentioned above, the corresponding absolute PLQYs were recorded in **Figure R17**.

Figure R17. (a-c) Calculation of PLQY of the aqueous solution of Au-Zn, Au-Zn/Ag, and Au-Zn/Ag/Tb NCs, respectively. (d) The statistics of recorded PLQY values of Au, Au-Zn, Au-Zn/Ag, and Au-Zn/Ag/Tb NCs, respectively.

Revisions:

SI, Page 8, Supplementary Fig. 2:

Figure R17 has been included as **Supplementary Fig. 2** in Supplementary Information.

SI, Page 6, Line 2-12:

“The PLQY of all samples in aqueous solution was measured and calculated on a QM8000 HORIBA spectrometer attached with an integrating sphere coating with a reflective BaSO₄ layer. For the detailed measurements of the PLQY of all samples, an Xe lamp with a fixed emission wavelength at 365 nm was used as the excitation source. Pure water support was measured first in the integrating sphere and used as blank references for the samples in an aqueous solution. The calculation of absolute

PLQY values was conducted on the built-in “FelixGX” software (version 4.9.0.10321). The values of PLQY were calculated by using following equation:

$PLQY = \frac{\int \lambda P(\lambda) d\lambda}{\int \lambda \{E(\lambda) - R(\lambda)\} d\lambda}$, where $E(\lambda)/h\nu$, $R(\lambda)/h\nu$, and $P(\lambda)/h\nu$ are the number of photons in the spectrum of excitation, reflectance, and emission, respectively.”

2. *The main reason is the lack of information on the structure and even composition of the “cluster”. Without this, the work remains highly speculative.*

Reply: We thank the reviewer for his/her careful critique. We totally agree with the reviewer that the information on the structure and even composition of a series NCs assemblies is insufficient (i.e., not atomically precise). 1) X-ray single crystal analysis is direct and of paramount importance to push our step in addressing the atomic-/molecular-precision structure. As for the water-soluble metal NCs, how to acquire X-ray-quality single crystals is still an open question and a big challenge. In our research system, there are strong electrostatic interactions between Zn^{2+} cations and deprotonated COO^- groups in the MPA (3-mercaptopropionic acid) ligands, combined with dipolar interactions between MPA ligands in water. In addition, the Zn^{2+} -induced assembly process of Au NCs has fast reaction kinetics and can be finished in several minutes. All these factors impede the formation of long-range-ordered packing to generate high-quality single crystals of Au NCs. We have tried to grow their high-quality single crystals but all our attempts failed to acquire X-ray-quality ones. 2) There are few successes in obtaining the single crystals of $-COOH$ functionalized metal NCs in some specialized systems, such as $Au_{102}(p-MBA)_{44}$ ($p-MBA = para$ -mercaptobenzoic acid, *Science* **2007**, 318, 430), and $Ag_{44}(p-MBA)_{30}$ (*Nature* **2013**, 501, 399). However, all of these successful single-crystal growths are based on the strong $\pi \cdots \pi$ interactions between neighboring $p-MBA$ ligands with high rigidity. While for aliphatic alkanes of MPA ligands with high flexibility in Au NCs, it is hard to form stable inter-ligand interactions to induce the generation of their single crystals. 3) Most of these crystalized metal NCs (e.g., $Au_{102}(p-MBA)_{44}$ and $Ag_{44}(p-MBA)_{30}$) have almost negligible PL properties. And, no special emphasis will be given to their atomically precise crystal structures for many interesting metal NCs assemblies when focusing on their high-efficiency PL properties (*Science* **2016**, 353, 571; *Nano Lett.* **2020**, 20, 2710; *J. Am. Chem. Soc.* **2021**, 143, 326). Therefore, to date, it generally can be regarded as a “trade-off” relationship between their excellent optical properties and atomically precise crystal structure. 4) Most importantly, the lack of atomically accurate structure and composition information of the series NCs does not affect our emphasis on the achievement of a near-unity PLQY of 99.5%, and on the mechanism revelation that efficient suppression of total structural vibrations and management of electron transfer dynamics (involved in multiple structural/compositional domains without the molecular/atomic precision) toward achieving remarkable enhancement in emission intensity.

3. In addition, the addition of metal ions makes the system a complex one, likely dominated by aggregation. Claims like (in the abstract): “Here, we report efficient suppression of the total structural motions (i.e., vibrations/rotations-dictated structure rigidity) and management of the electron transfer dynamics (i.e., pathways and rates of the radiative relaxation) to boost a near-unity absolute PLQY, by decorating progressive addition of cations.” are then difficult to judge (or prove), as many different effects may compete.

Reply: Thank you for this insightful comment. We are sorry that we didn’t articulate well the structural evolution and the underlying photophysical mechanisms.

Firstly, the characterization such as TEM and DLS proved that as-synthesized a series of NCs were indeed dominated by aggregation. As shown in **Figure R18a**, the individual Au NCs feature an average diameter of 2.47 ± 0.44 nm with excellent monodispersity. After adding Zn^{2+} , Ag^+ , and Tb^{3+} into the individual Au NCs, a series of NCs show irregular aggregation. The size of a series of NCs is highly consistent with the results of DLS (i.e., 230, 270, and 360 nm for Au-Zn, Au-Zn/Ag, and Au-Zn/Ag/Tb NCs, respectively). High-resolution transmission TEM images show that Au-Zn, Au-Zn/Ag, and Au-Zn/Ag/Tb NCs are assembled by the individual NCs with similar average diameters of 2.50 ± 0.44 nm, 2.52 ± 0.42 , and 2.52 ± 0.50 nm, respectively (**Figure R18**).

Secondly, to prove that stepwise metal cation addition can effectively suppress the total structural motions, we have carried out a series of temperature-dependent PL and fs-TA measurements. As shown in **Figure R19**, the fitting results of the temperature-dependent PL FWHM of the three kinds of NCs reveal that the coupling strength of electrons-high-frequency vibration related to the surface-interface motif and ligand is indeed getting much smaller (40.2, 30.5, and 14.4 meV for Au-Zn, Au-Zn/Ag, and Au-Zn/Ag/Tb NCs), confirming that the vibration of surface-interface motif and ligand is gradually suppressed after cation additions. In the fs-TA fitting curves (**Figure R20 g-i**), it can be seen that the time component of core-directed structural vibration gradually increases (i.e., 3.1, 3.8, and 5.6 ps for Au-Zn, Au-Zn/Ag, and Au-Zn/Ag/Tb NCs, respectively), indicating that the vibration of the metal core is greatly suppressed after cation additions.

Finally, to gain further insights into the effect of stepwise metal cation addition on electron transfer dynamics, a series of characterizations, including PLE and fs-TA spectra, have been performed. The appearance of the electron transfer process is supported by their corresponding excitation spectra (**Figure R21**), where the contribution of the excitation of staple motifs gradually increases with the subsequent addition of the metal cations. In the fs-TA fitting curves (**Figure R20 g-i**), the time component of dozens of picoseconds processes (i.e., 40, 27, and 12 ps) relate to the electron transfer process from the excited state of staple motifs (the electron donor) to the metal core (the electron acceptor). The gradual decrease of this time component manifests that the electron transfer process is getting faster with the sequential addition of metal cations. In addition, the S_1 energy levels of core in Au-Zn, Au-Zn/Ag, and Au-Zn/Ag/Tb NCs are calculated to be 20.0×10^3 , 20.4×10^3 , and 20.4×10^3 cm^{-1} , respectively.

The S_1 energy level position of the staple motif in Au-Zn, Au-Zn/Ag, and Au-Zn/Ag/Tb NCs can be estimated to be 28.2×10^3 , 29.7×10^3 , and $29.1 \times 10^3 \text{ cm}^{-1}$. The inherent 5D_3 energy level of rare earth ion Tb^{3+} is $26.3 \times 10^3 \text{ cm}^{-1}$ (*Chem. Rev.* **2022**, 122, 5519-5603; *Light: Sci. Appl.* **2016**, 5, e16066), which is located between the S_1 level of the staple motif and the S_1 level of the metal core and can serve as a bridge to promote the electron transfer process.

Combine the above data and analyses, we thus claim that “efficient suppression of the total structural motions (i.e., vibrations/rotations-dictated structure rigidity) and management of the electron transfer dynamics (i.e., pathways and rates of the radiative relaxation) to boost a near-unity absolute PLQY, by decorating progressive addition of cations”.

Figure R18. (a-d) TEM images of Au, Au-Zn, Au-Zn/Ag, and Au-Zn/Ag/Tb NCs, respectively. The insets are amplified images of the corresponding samples.

Figure R19. (a-c) Temperature-dependent PL map of Au-Zn, Au-Zn/Ag, and Au-Zn/Ag/Tb NCs. The excitation source is 365 nm light and the temperature range is 70–290 K with a temperature interval of 20 K. (d-f) The PL FWHM of Au-Zn, Au-Zn/Ag, and Au-Zn/Ag/Tb NCs as a function of temperature and their corresponding fitting according to the weak electron-vibration coupling approximation model.

Figure R20. Comparison of femtosecond-TA spectra and their electron dynamics of serial NCs. (a-c) The fs-TA map pumped at 3.40 eV (365 nm). (d-f) TA profile at different time delays. (g-i) The ESA kinetic decay and fitted residuals within 500 ps (from left to right: Au-Zn, Au-Zn/Ag, and Au-Zn/Ag/Tb NCs).

Figure R21. (a-c) Excitation spectra of Au-Zn, Au-Zn/Ag, and Au-Zn/Ag/Tb NCs.

4. Concerning the cluster structure/composition: The authors seem to imply, based on mass spectrometry, that the cluster is very small, four to five metal atoms only. (The masses observed could actually be fragments!) Light scattering on the other hand

implies something bigger (a few nm). In any case, the authors are not sure. (What about analysis by gel electrophoresis). The authors are also not sure if the sample is monodisperse, which further complicates the interpretation.

Reply: Thank you for this insightful comment. To further identify structure/composition, we have carried out the electrospray ionization mass spectrometry (ESI-MS, the “soft” MS technique) analysis of serial Au NCs in both positive and negative-ion modes. As shown in **Figure R22**, it can be observed that the mass signals of intact NCs are hard to get for all three Au NCs regardless of the measuring mode. It can be rationally explained from the following insights. (i) excess small ions (e.g., Na⁺ and OH⁻) at the alkaline conditions will seriously disturb the ESI-MS signals; (ii) To maintain the stability and superior PL properties of NCs, excess MPA ligand (~ 40 folds higher in molar concentration) are typically indispensable. The excess self-charged COO⁻ of MPA ligands under alkaline conditions can significantly interfere with the mass signals of intact NCs. Even if we have attempted to purify these NCs several times, there are still no mass signals detected for NCs. (iii) The as-synthesized Au NCs are organized in an assembly manner and not easily dispersed in solution, which further impedes their ionization in ESI-MS measurements. (iv) Excess metal counter ions (i.e., Zn²⁺) were added to induce the self-assembly of Au NCs, which can seriously disturb the ESI-MS signals (*Angew. Chem. Int. Ed.* **2019**, 58, 11967). However, it should be noted that metal NCs in a solution state are very dynamic due to the underlying absorption and desorption of external ligands on the surface of metal NCs. For our NCs assembly systems with multiple metal cation additions, quantitatively acquiring their chemical formulas gets more difficult. Therefore, we have implemented a series of MALDI-TOF measurements with varied laser power to identify the composition of as-synthesized NCs assembly. It is worth noting that the laser power should be large enough (> 70% of the maximum power) to isolate the main component from its assembly. As shown in **Figure R23**, with the increase of laser power from 70% to 95%, the mass signal at 1147.3 (Au-Zn) and 1277.2 (Au-Zn/Ag, and Au-Zn/Ag/Tb) Da gradually weakened while the mass signals in front were accordingly enhanced. Therefore, the mass signals at 1147.3 (Au-Zn) and 1277.2 (Au-Zn/Ag, and Au-Zn/Ag/Tb) Da are assigned to the compositions of [Au₄(MPA)₃+2Na-2H], [Au₄Ag(MPA)₃+3Na-3H], respectively, which are the relatively stable structural building blocks in the superstructure of metal NCs assemblies.

In addition, DLS shows that stepwise adding Zn²⁺, Ag⁺, and Tb³⁺ to Au NCs significantly increases the average size from 2.47 to 230, 270, and 360 nm, suggesting that the NCs have been assembled after the addition of Zn²⁺ by strong electrostatic interactions between Zn²⁺ and deprotonated COO⁻ groups in MPA ligands (**Figure R24**). TEM images also show that Au-Zn, Au-Zn/Ag, and Au-Zn/Ag/Tb are assembled by the individual NCs with similar average diameters of 2.50 ± 0.44 nm, 2.52 ± 0.42, and 2.52 ± 0.50 nm (**Figure R18**).

As suggested by the reviewer, we have conducted native polyacrylamide gel electrophoresis (PAGE, 30%) to separate and purify the serial target NCs and further attempt to reveal their structure/composition. However, the serial target NCs were

found to be blocked into native polyacrylamide gel and cannot be separated due to the assembly nature of serial target NCs and the high sensitivity of target NCs to external electric fields (**Figure R25**). Therefore, decoding the structure/composition of serial aqueous NCs through the PAGE method is not accessible.

To sum up, as stepwise adding Zn^{2+} , Ag^+ , and Tb^{3+} , the NCs will self-assemble by electrostatic interactions, along with the average size evolution from 2.47 to 230, 270, and 360 nm, concurrent with the slight composition evolution of the NCs building blocks from $[\text{Au}_4(\text{MPA})_3]$ to $[\text{Au}_4\text{Ag}(\text{MPA})_3]$, in average.

Figure R22. (a-c) ESI-MS spectra of Au-Zn, Au-Zn/Ag, and Au-Zn/Ag/Tb NCs assemblies measured in positive-ion and negative-ion modes.

Figure R23. (a-c) Positive MALDI-TOF mass spectra of Au-Zn, Au-Zn/Ag, and Au-Zn/Ag/Tb NCs assemblies recorded by changing laser power.

Figure R24. DLS measurements of a series of NCs before and after Zn^{2+} , Ag^+ , and Tb^{3+} addition.

Figure R25. (a-c) Digital photos of the PAGE gels of the as-synthesized Au-Zn, Au-Zn/Ag, Au-Zn/Ag/Tb NCs under visible (left) and UV (right) light.

Revisions:

Page 7, Line 19 to Page 8, Line 1:

“To push our step in addressing the composition of as-synthesized NCs, we conducted the matrix-assisted laser desorption ionization-time-of-flight mass. As shown in **Supplementary Fig. 6**, the mass signals at 1147.3 (Au-Zn) and 1277.2 (Au-Zn/Ag, and Au-Zn/Ag/Tb) Da are assigned to the compositions of $[\text{Au}_4(\text{MPA})_3+2\text{Na}-2\text{H}]$, $[\text{Au}_4\text{Ag}(\text{MPA})_3+3\text{Na}-3\text{H}]$, respectively.”

5. *In the following the authors analyse their data in the frame of electron–phonon coupling. They talk about optical and acoustic phonons and apply models developed in the field of solid state physics. Talking about (optical and acoustic) phonons for a cluster that has 4 atoms in the core does absolutely not make sense. (Even for a cluster with 1-2 nm core it is not appropriate.*

Reply: We greatly appreciate the reviewer for the professional and constructive comment, which inspired us to revisit and deepen the related knowledge. Electron-phonon coupling is one of the most momentous fundamental interactions in solid-state physics systems. The interaction of electrons with phonons (atomic and lattice vibrations) in solids with long-range order of arrangement of atoms or molecules is of central importance in chemistry and physics, defining carrier transport properties,

superconducting critical temperatures, the rate of nonradiative recombination, the reaction pathway of ultrafast charge and energy transfer. In recent years, with the in-depth exploration of electron-phonon coupling, in addition to solid-state physics systems, researchers have found that electron-phonon coupling also plays a vital role in metal NCs or quantum dots composed of tens to hundreds of atoms. Electron-phonon coupling in metal NCs has been widely studied. For example, in 2018, Jin et al. investigated experimentally by ultrafast time-resolved optical spectroscopy the acoustic response of atomically defined ligand-protected metal clusters $\text{Au}_n(\text{SR})_m$ with a number n of atoms ranging from 10 to 102 (0.5-1.5 nm diameter range) (*Nano Lett.* **2018**, 18, 6842). In 2021, they found for the first time that the Au core phonon (optical phonon) was exhibited by significantly suppressing the Au-S shell phonon in Au_{25} nanoclusters solid film. (*J. Phys. Chem. Lett.* **2021**, 12, 1690). In 2022 and 2023, the effects of ligand type on the electron-phonon interaction and luminescence of $\text{Au}_{25}(\text{SR})_{18}$ and $\text{Au}_{52}(\text{SR})_{32}$ nanoclusters were also investigated (*ACS Nano* **2022**, 16, 18448; *J. Am. Chem. Soc.* **2023**, 145, 26328). In addition, different heteroatoms and different doping sites in $\text{Au}_{25}(\text{SR})_{18}$ led to the change of electron-phonon coupling mode, that is, the strong coupling mode dominated by shell doping is transformed into the weak coupling mode dominated by core doping (*J. Am. Chem. Soc.* **2023**, 145, 19969). However, in 2023, C. Crowther et al. explored the vibrational structure of serial atomically precise cadmium selenide quantum dots ($\text{Cd}_{35}\text{Se}_{20}\text{X}_{30}\text{L}_{30}$, $\text{Cd}_{56}\text{Se}_{35}\text{X}_{42}\text{L}_{42}$, and $\text{Cd}_{84}\text{Se}_{56}\text{X}_{56}\text{L}_{56}$). As the size of the quantum dots gradually increases, it was found that the transition from molecular vibrations to phonons in atomically precise cadmium selenide quantum dots (*J. Am. Chem. Soc.* **2016**, 138, 16754). In 2024, Jin et al. investigated the effects of electron-vibration coupling on the photoluminescence of isomeric $\text{Au}_{28}(\text{S-c-C}_6\text{H}_{11})_{20}$ nanoclusters. The isomer of $\text{Au}_{28\text{i}}$ has a high-frequency vibrational mode in coupling with the band gap electronic transition. In contrast, the coupled vibration of the $\text{Au}_{28\text{ii}}$ isomer is low-frequency (*ACS Nano* **2024**, 18, 21534). The above studies show that the size and structure of the material greatly affect the electron coupling mode (e.g., electron-phonon or electron-vibration).

However, as the reviewer mentioned, for the as-synthesized metal NCs with only four atoms and disordered aggregates without periodic arrangement, the coupling mode is better described as the coupling of vibration and electronic transitions, instead of the coupling of phonon and electronic transitions. Therefore, it is not appropriate to discuss the phonons in the cluster. We totally share the viewpoints in the reviewer's comments, we thus have modified the corresponding statements (e.g., "optical phonons" and "acoustic phonons" were replaced by the wordings of "high-frequency vibrations" and "low-frequency vibrations", respectively) in the revised manuscript to make the related expressions more rigorous.

Revisions:

Page 2, Line 9-11:

“...the low-frequency vibration of the metal core progressively decreases from 144.0, 55.2 to 40.0 cm⁻¹, and the coupling strength of electrons-high-frequency vibration related to surface motifs...”

Page 3, Line 11-13:

“On this basis, low- and high-frequency vibrations can be induced in the metal core and staple motifs, respectively^{12, 13}. The coupling of multiple vibrations and excited-state electrons...”

Page 4, Line 6:

“...electron-vibration coupling strength²³⁻²⁶.”

Page 4, Line 18:

“...due to the changed electron-vibration interactions.”

Page 11, Line 17-18:

“...indicating the strong electron-vibration interaction.”

Page 11, Line 23-24:

“...suggesting that the electron-vibration interaction-induced non-radiative relaxations.”

Page 12, Line 5-7:

“...the activation energies that the low-vibration modes coupled with the emission peaks of the Au-Zn, Au-Zn/Ag, and Au-Zn/Ag/Tb NCs are 18.0, 6.90, and 5.00 meV, respectively. The corresponding vibration-coupled frequencies...”

Page 12, Line 13-15:

“...and low-frequency vibration coupling mode of the NCs provide solid proof that the PL of these NCs originates from the metal core. Moreover, the gradual decrease in the vibration-coupled frequency confirms...”

Page 12, Line 24 to Page 13, Line 12:

“Interestingly, the temperature-dependent PL FWHM of all NCs in the high-temperature (> 150 K) region follows the linear relationship, indicating the common weak electron-vibration coupling approximation mode is applicable²⁸:

$$\Gamma(T) = \Gamma_0 + \gamma_{LF}T + \gamma_{HF} \frac{1}{\exp\left(\frac{E_{HF}}{k_B T}\right) - 1} \quad (4)$$

where Γ_0 is the FWHM of NCs at 0 K, γ_{LF} and γ_{HF} are the coupling coefficients of an electron with low-frequency vibration (LF) and high-frequency vibration (HF), respectively. E_{HF} is the average energy of the HF vibration. Since we only focus on the coupling of electron-high-frequency vibration caused by staple motif and tail ligand at high temperature, therefore, “ $\gamma_{LF}T$ ” representing the electron-low-

frequency vibration is not considered. Accordingly, the weak electron-vibration coupling approximation model is simplified to equation (5)²⁸:

$$\Gamma(T) = \Gamma_0 + \gamma_{\text{HF}} \frac{1}{\exp\left(\frac{E_{\text{HF}}}{k_B T}\right) - 1} \quad (5)$$

It is worth noting that the coupling coefficients of the electron with high-frequency vibrations are significantly reduced with sequential cation addition ($\gamma_{\text{HF}} = 40.2$, 30.5, and 14.4 meV for Au-Zn, Au-Zn/Ag, and Au-Zn/Ag/Tb, respectively). Therefore, the addition of metal cations can also be able to suppress the coupling of excited electrons and high-frequency vibration in the surface and interface in metal NCs (Fig. 3d).”

Page 13, Line 19-20:

“...their corresponding fitting according to the weak electron-vibration coupling approximation model...”